# A cellular and regulatory map of the cholinergic nervous system of *C. elegans*

Laura Pereira[1,2,3]*, Paschalis Kratsios[1,2,3†], Esther Serrano-Saiz[1,2,3†], Hila Sheftel[4], Avi E Mayo[4], David H Hall[5], John G White[6], Brigitte LeBoeuf[7], L Rene Garcia[7,8], Uri Alon[4], Oliver Hobert[1,2,3]*

[1]Department of Biological Sciences, Columbia University, New York, United States; [2]Department of Biochemistry and Molecular Biophysics, Columbia University, New York, United States; [3]Howard Hughes Medical Institute, Columbia University, New York, United States; [4]Department of Molecular Cell Biology, Weizmann Institute of Science, Rehovot, Israel; [5]Department of Neuroscience, Albert Einstein College of Medicine, New York, United States; [6]MRC Laboratory of Molecular Biology, Cambridge, United Kingdom; [7]Department of Biology, Texas A&M University, College Station, United States; [8]Howard Hughes Medical Institute, Texas A&M University, College Station, United States

**Abstract** Nervous system maps are of critical importance for understanding how nervous systems develop and function. We systematically map here all cholinergic neuron types in the male and hermaphrodite *C. elegans* nervous system. We find that acetylcholine (ACh) is the most broadly used neurotransmitter and we analyze its usage relative to other neurotransmitters within the context of the entire connectome and within specific network motifs embedded in the connectome. We reveal several dynamic aspects of cholinergic neurotransmitter identity, including a sexually dimorphic glutamatergic to cholinergic neurotransmitter switch in a sex-shared interneuron. An expression pattern analysis of ACh-gated anion channels furthermore suggests that ACh may also operate very broadly as an inhibitory neurotransmitter. As a first application of this comprehensive neurotransmitter map, we identify transcriptional regulatory mechanisms that control cholinergic neurotransmitter identity and cholinergic circuit assembly.

*For correspondence: pereira. lau@gmail.com (LP); or38@ columbia.edu (OH)

†These authors contributed equally to this work

## Introduction

Nervous system maps that describe a wide range of distinct structural and molecular parameters are essential for an understanding of nervous system development and function. Tremendous efforts have been and are being made to map connectomes (*Bargmann and Marder, 2013*; *Plaza et al., 2014*). Connectomes now exist for small anatomic regions of mouse and fly brains (*Helmstaedter et al., 2013*; *Kasthuri et al., 2015*; *Takemura et al., 2013*), but the only complete, system-wide connectome remains that of the nematode *Caenorhabditis elegans*, both in its hermaphroditic and male form (*Albertson and Thomson, 1976*; *Jarrell et al., 2012*; *White et al., 1986*). However, these anatomical maps are incomplete without the elucidation of chemical maps that describe the synaptically released neurotransmitters through which anatomically connected neurons communicate with one another. But even in *C. elegans*, let alone other organisms, there have so far been only limited efforts to precisely map neurotransmitter identities on a system-wide level with single-neuron resolution. In *C. elegans*, a combination of direct staining methods and expression analysis of neurotransmitter-specific enzymes and transporters have defined the probably complete complement of GABAergic, glutamatergic and aminergic neurotransmitter systems.

**eLife digest** To better understand the nervous system—the most complex of all the body's organs—scientists have begun to painstakingly map its many features. These maps can then be used as a basis for understanding how the nervous system develops and works.

Researchers have mapped the connections – called synapses – between all the nerve cells in the nervous system of a simple worm called *Caenorhabditis elegans*. Cells communicate by releasing chemicals called neurotransmitters across the synapses, but it is not fully known which types of neurotransmitters are released across each of the synapses in *C. elegans*.

Now, Pereira et al. have mapped all worm nerve cells that use a neurotransmitter called acetylcholine by fluorescently marking proteins that synthesize and transport the neurotransmitter. This map revealed that 52 of the 118 types of nerve cells in the worm use acetylcholine, making it the most widely used neurotransmitter. This information was then combined with the findings of previous work that investigated which nerve cells release some other types of neurotransmitters. The combined data mean that it is now known which neurotransmitter is used for signaling by over 90% of the nerve cells in *C. elegans*.

Using the map, Pereira et al. found that some neurons release different neurotransmitters in the different sexes of the worm. Additionally, the experiments revealed a set of proteins that cause the nerve cells to produce acetylcholine. Some of these proteins affect the fates of connected nerve cells. Overall, this information will allow scientists to more precisely manipulate specific cells or groups of cells in the worm nervous system to investigate how the nervous system develops and is regulated.

Specifically, out of the 118 anatomically distinct neuron classes in the hermaphrodite (amounting to a total of 302 neurons), six classes (26 neurons) are GABAergic (*McIntire et al., 1993*), 38 are glutamatergic (78 neurons) (*Serrano-Saiz et al., 2013*) and 13 (26 neurons) are aminergic (i.e. serotonergic, dopaminergic, *etc.; Chase and Koelle, 2007*).

One prominent neurotransmitter system – the cholinergic system – has not been completely mapped. Antibody staining against the vesicular acetylcholine (ACh) transporter, VAChT (encoded by *unc-17*) and the ACh-synthesizing choline acetyltransferase ChAT (encoded by *cha-1*) revealed the cholinergic identity of a number of neurons in the nervous system (*Alfonso et al., 1993*; *Duerr et al., 2008*). However, due to the synaptic localization of the VAChT and ChAT proteins, expression could only be unambiguously assigned to about one dozen neuron classes, mostly in the ventral nerve cord and a few isolated head and tail neurons (see *Table 1* for a summary of previous studies on cholinergic neuron identity). The authors of these previous studies explicitly noted that many additional VAChT/ChAT-expressing neuron classes await identification (*Duerr et al., 2008*). Ensuing studies using reporter genes that capture *cis*-regulatory elements of parts of the *unc-17*/VAChT locus identified the cholinergic identity of a few additional neuron classes (*Table 1*), but the extent to which ACh is used in the nervous system has remained unclear. In the male nervous system, composed of 23 additional neuron classes, neurotransmitter identities are even less well defined (not just ACh, but other systems as well). Here, we map the usage of ACh in both the hermaphrodite and male nervous systems. We show that ACh is the most broadly used neurotransmitter in the *C. elegans* nervous system, employed by more than half of all neurons.

The tremendous benefits of a neurotransmitter map include the ability to precisely dissect and understand neuronal circuit function. For example, knowledge of the cholinergic identity of the AIY interneuron (*Altun-Gultekin et al., 2001*) helped to define the two distinct behavioral outputs of AIY, one controlled via an ACh-mediated activation of the RIB interneuron and another controlled by ACh-mediated inhibition of the AIZ interneuron, via an ACh-gated chloride channel (*Li et al., 2014*). The cholinergic neurotransmitter map presented here will provide a resource to further functionally dissect circuit function in the *C. elegans* nervous system.

Since neurotransmitter identity represents a key feature of a neuron, the knowledge of the cholinergic identity provides a resource for studying how a neuron adopts its specific fate during development. For example, the assignment of glutamatergic identity to a host of distinct *C. elegans* neurons

**Table 1.** Cholinergic neurons in the hermaphrodite.

| Neuron type | Neuron class | VAChT/ChAT[1] | ChT [2] | AChE [3] | Co-transmitter | Previous ID [4] |
|---|---|---|---|---|---|---|
| Sensory neuron (9 classes) | ADF L/R | ++ | ++ | | Serotonin | no |
| | ALN L/R | ++ | ++ | | | yes [5] |
| | ASJ L/R | + | ++ | | | no |
| | AWB L/R | ++ | ++ | | | no |
| | IL2 D/V L/R | +++ | +++ | ace-3/4 | | yes [6] |
| | PLN L/R | ++ | ++ | | | yes [5] |
| | URA D/V L/R | +++ | ++ | ace-3/4 | | yes [6] |
| | URB L/R | ++ | + | ace-3/4 | | yes [6] |
| | URX L/R | ++ | ++ | ace-3/4 | | no |
| Interneuron (19 classes) | AIA L/R | +++ | ++ | ace-3/4 | | yes [7] |
| | AIN L/R | ++ | +++ | | | yes [6] |
| | AIY L/R | +++ | +++ | | | yes [7] |
| | AVA L/R | +++ | ++ | ace-2 | | no |
| | AVB L/R | +++ | ++ | ace-2 | | no |
| | AVD L/R | +++ | ++ | ace-2 | | no |
| | AVE L/R | +++ | ++ | ace-2 | | no |
| | AVG | + | | | | no |
| | DVA | +++ | +++ | ace-2; ace-3/4 | | no [8] |
| | PVC L/R | +++ | + | | | yes [5] |
| | PVN L/R | ++ | | | | no |
| | PVP L/R | +++ | ++ | | | yes [5] |
| | RIB L/R | (+)* | +++ | | | no |
| | RIF L/R | ++ | ++ | | | no |
| | RIH | +++ | ++ | ace-2; ace-3/4 | Serotonin | no |
| | RIR | ++ | ++ | | | no |
| | RIV L/R | ++ | ++ | ace-3/4 | | no |
| | SAA D/V L/R | ++ | | | | no |
| | SDQ L/R | ++ | ++ | | | yes [5] |
| Motor neuron (17 classes) | AS1-11 | ++ | ++ | ace-2 | | yes [5] |
| | DA1-9 | ++ | ++ | ace-2 | | yes [5] |
| | DB1-7 | ++ | ++ | ace-2 | | yes [5] |
| | HSN L/R | ++ | | | Serotonin | yes [5] |
| | PDA | ++ | ++ | ace-2; ace-3/4 | | no |
| | PDB | ++ | ++ | | | no |
| | RMD D/V L/R | +++ | +++ | ace-3/4 | | yes [5] |
| | RMF L/R | ++ | ++ | | | no |
| | RMH L/R | ++ | ++ | | | no |
| | SAB D V L/R | ++ | ++ | | | yes [9] |
| | SIA D/V L/R | +++ | ++ | ace-3/4 | | no |
| | SIB D/V L/R | +++ | ++ | | | no |
| | SMB D/V L/R | +++ | ++ | | | no |
| | SMD D/V L/R | +++ | +++ | ace-3/4 | | no [10] |

*Table 1 continued on next page*

*Table 1 continued*

| Neuron type | Neuron class | VAChT/ChAT[1] | ChT[2] | AChE[3] | Co-transmitter | Previous ID[4] |
|---|---|---|---|---|---|---|
| Motor neuron (17 classes) | VA1-12 | ++ | ++ | *ace-2* | | yes[5] |
| | VB1-11 | ++ | ++ | *ace-2* | | yes[5] |
| | VC1-3 VC6 | ++ | ++ | | | yes[5] |
| | VC4-5 | ++ | | | Serotonin | yes[5] |
| | Pharyngeal | | | | | |
| Polymodal (7 classes) | I1 L/R | ++ | | | | no |
| | I3 | ++ | | | | no |
| | MC L/R | ++ | | | | yes[11] |
| | M1 | ++ | | | | no |
| | M2 L/R | ++ | | | | no |
| | M4 | +++ | +++ | *ace-2* | | no |
| | M5 | +++ | +++ | | | no |
| | *unc-17(+)*: 52 classes, 159 neurons | | | | | |

See the legend to **Figure 2A** and **Table 2** for notes on neuron classification. Data for the male nervous system is shown in **Table 5**. '+' indicate relative expression levels. See **Figure 1** for images.

*Expression of *cho-1* in the RIB neurons is strong but *unc-17* expression is, at best, very dim.

[1]Gray shading indicates *unc-17/cha-1* (VAChT/ChAT) expression as assessed by fosmid reporter and antibody staining.

[2]Gray shading indicates *cho-1* (ChT) expression as assessed by fosmid reporters.

[3]Gray shading indicates reporters expression of one of the *C. elegans ace* (AChE) genes.

[4]Previously identified as a cholinergic neuron: 'yes' – see indicated references. 'no' - newly identified in this study. Only published data is considered, personal communications in **Rand and Nonet (1997)** were not taken into consideration.

[5]**Duerr et al. (2008)**.

[6]**Zhang et al. (2014)**.

[7]**Altun-Gultekin et al. (2001)**.

[8]Previously proposed to be DVC (**Duerr et al., 2008**) but based on position and markers reassigned to DVA.

[9]**Zhao and Nonet (2000)**.

[10]Based on our identification as SMB as cholinergic, **Kim et al. (2015)** demonstrated that *lim-4* controls SMB cholinergic identity (see also **Table 6**).

[11]**Raizen et al. (1995)**.

has enabled us to define phylogenetically conserved regulatory features of glutamatergic neuron differentiation (**Serrano-Saiz et al., 2013**). Moreover, the long-known cholinergic identity of ventral cord motor neurons provided an entry point to study how their terminal differentiation is controlled (**Kratsios et al., 2011**; **2015**). Previous studies describing the mechanism of cholinergic identity regulation have pointed to a modular control system in which neuron-type specific combinations of transcription factors turn on cholinergic pathway genes (**Altun-Gultekin et al., 2001**; **Kratsios et al., 2011**; **Zhang et al., 2014**). Since previous studies only examined a relatively small number of neurons, the problem of cholinergic identity regulation has not yet encompassed a circuit level analysis. Through a genetic screen and a candidate gene approach we reveal common themes in the form of circuit-associated transcription factors that control the identity of all neurons within defined circuits or circuit-associated network motifs. Taken together, we anticipate that neurotransmitter maps like those provided here represent an invaluable resource for the *C. elegans* community that will serve as a high-resolution starting point for various types of behavioral and developmental analyses.

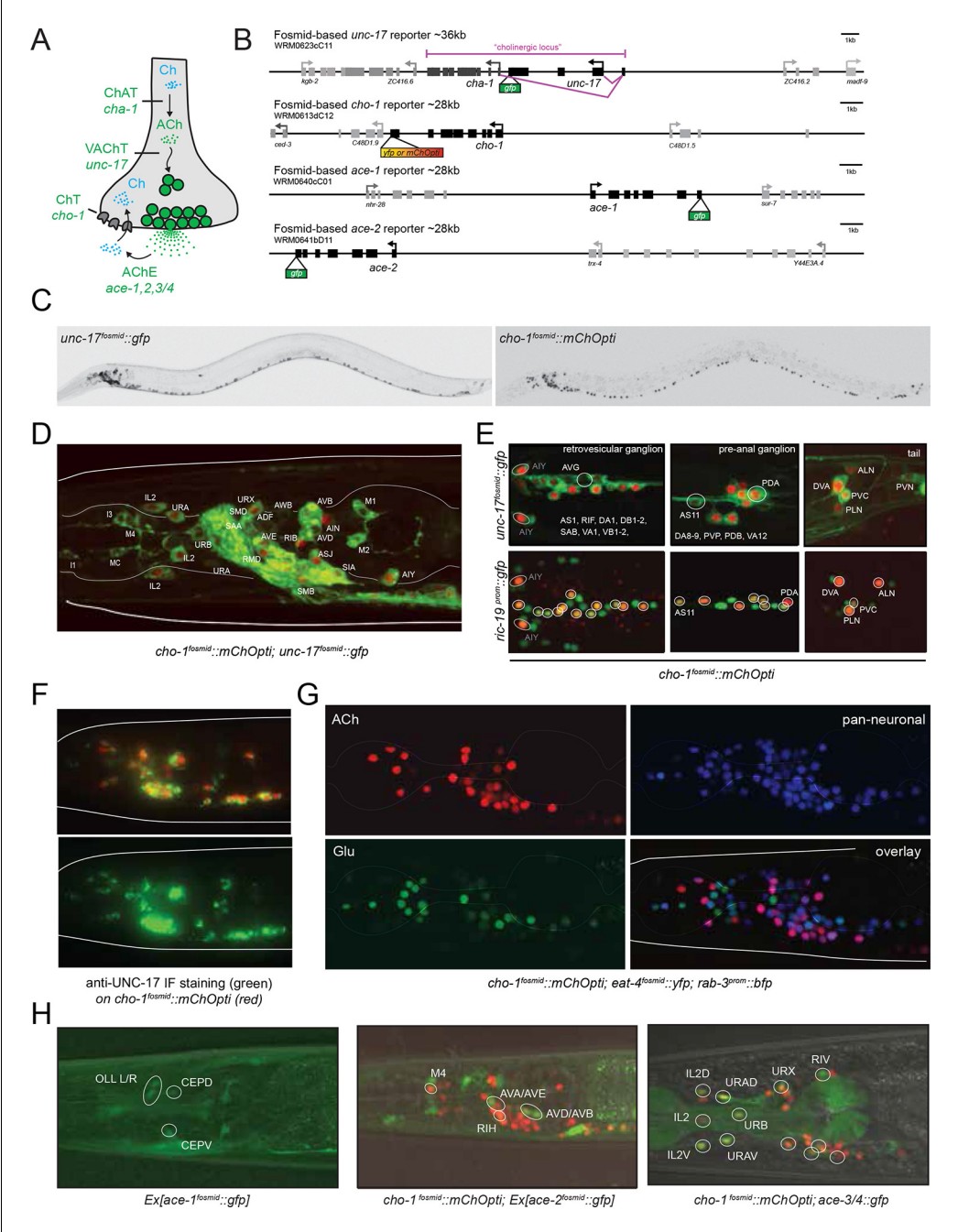

**Figure 1.** Expression of cholinergic pathway genes in the adult *C. elegans* hermaphrodite. (**A**) Cholinergic pathway genes. Ch = choline; ACh = acetylcholine; ChAT = choline acetyltransferase; VAChT = vesicular ACh transporter, AChE = ACh esterase, ChT = choline transporter. (**B**) Fosmid reporters used in this study. The *unc-17* fosmid reporter was kindly provided by the TransgeneOme project (*Sarov et al., 2012*). It was previously reported that the expression of *unc-17/VAChT* and *cha-1/ChAT* overlap completely (*Mathews et al., 2015*). (**C**) *unc-17* and *cho-1* fosmid reporter expression in an L4 hermaphrodite. The fluorescent reporter inserted into the *cho-1* locus is targeted to the nucleus (see Materials and methods), while the fluorescent reporter inserted into the *unc-17* locus is fused directly to the *unc-17* gene (resulting in cytoplasmic localization). (**D, E**) *unc-17* and *cho-1* fosmid reporter expression in head (**D**), retrovesicular ganglion and tail ganglia (**E**). In (**E**) bottom panels, neurons are labeled with a green pan-neuronal marker, *ric-19*. Transgenes: *otIs576* = *unc-17* fosmid reporter; *otIs544* = *cho-1* fosmid reporter, *otIs380* = *ric-19* reporter (*Stefanakis et al., 2015*). (**F**) Immunofluorescent staining for endogenous UNC-17 protein of *unc-104(e1265)* animals that express the *cho-1* fosmid reporter transgene *otIs544*. (**G**) Co-labeling cholinergic (*cho-1/ChT*-positive) and glutamatergic (*eat-4/VGLUT*-positive) neurons illustrate no overlap in neurotransmitter ACh and Glu expression, and co-labeling with pan-neuronal marker *rab-3* illustrates that most neurons now have a neurotransmitter assignment. Transgenes: *otIs544* = *cho-1* fosmid reporter, *otIs388* = *eat-4* fosmid reporter (*Serrano-Saiz et al., 2013*), *otIs355* = *rab-3* reporter. (**H**) *ace/AChE* genes are expressed in a subset of cholinergic neurons and in non-cholinergic neurons. *ace-1 fosmid* reporter expression in head neurons (left panel). *ace-2* fosmid reporter

*Figure 1 continued on next page*

*Figure 1 continued*
expression in head neurons together with *cho-1* fosmid reporter (middle panel). *ace-3/4* reporter expression together with *cho-1* fosmid reporter in head neurons (right panel). Transgenes: *otEx4435 = ace-1* fosmid reporter; *otEx4431 = ace-2* fosmid reporter; *fpIs1 = ace-3/4* transcriptional reporter.
The following figure supplements are available for figure 1:

**Figure supplement 1.** Neuronal cell identification.
**Figure supplement 2.** Neurotransmitter identity of pharyngeal neurons.
**Figure supplement 3.** Expression of *unc-17* and *cho-1* fosmid reporters in the male tail.

## Results and discussion

### Defining cholinergic neurons

Cholinergic neurotransmitter identity is defined by the expression of the enzyme choline acetyltransferase (ChAT; encoded by *cha-1* in *C. elegans*) and the vesicular ACh transporter (VAChT; encoded by *unc-17* in *C. elegans*); see *Figure 1A* for a description of the cholinergic pathway genes. Co-expression of these two genes is ensured via their organization into an operon-like structure called the cholinergic locus (*Figure 1B*). This operon-like organization is conserved from invertebrates to vertebrates (*Eiden, 1998*). Other possible diagnostic features of cholinergic neurons often used in vertebrates are the expression of the enzyme that breaks down ACh, acetylcholinesterase (AChE/ *ace*; four genes in *C. elegans*; [*Arpagaus et al., 1998*]) and the reuptake transporter of the breakdown product choline (ChT; encoded by *cho-1* in *C. elegans* [*Okuda et al., 2000*]). Whether these genes are expressed in all cholinergic neurons and/or restricted to all cholinergic neurons is, however, unclear.

To define cholinergic neuron types, we generated transgenic lines expressing fosmid based reporters for the *unc-17* and *cha-1* locus, the *cho-1* locus and several *ace* genes (*Figure 1B*). Fosmids contain 30–40 kb genomic sequences, including genes upstream and downstream of the gene of interest and usually contain all *cis*-regulatory information involved in regulating expression of a specific gene. Differently colored fluorescent proteins were used to assess the relative overlap of these genes to one another (*Figure 1C–E*). The fosmid lines that monitor *cho-1* and *ace-1/-2* expression are nuclear localized reporters, in which the fluorescent tag is separated from the respective genomic locus by an SL2 trans-splicing event and targeted to the nucleus (see Materials and methods). The fosmid line for the *unc-17* locus is, in contrast, a direct fusion of *gfp* to the *unc-17* gene, thereby revealing the subcellular localization of *unc-17*.

Multiple lines for each reporter transgene were analyzed and no differences between lines were found (for example, green and red fluorescent signals from a *cho-1^fosmid^::mCherry* transgenic line and an independent *cho-1^fosmid^::yfp* transgenic line perfectly overlap; data not shown). Preliminary neuron identifications were done based on cell position and axonal projections. These identifications were then confirmed for each neuron by crossing the *unc-17* and/or *cho-1* fosmid reporter strains with a differently colored reporter with a known, neuron type-specific expression pattern (*Figure 1— figure supplement 1*; see also Materials and methods). Furthermore, we validated *unc-17* fosmid reporter expression by immunofluorescent staining with an antibody generated against the UNC-17 protein (*Duerr et al., 2008*). As previously noted, the punctate localization of UNC-17 protein, as detected with the UNC-17 anti-serum, limits the ability to reliably identify cells in the absence of markers (*Duerr et al., 2008*). However, immunostaining for UNC-17 in combination with the nuclear localization of *cho-1* fosmid reporter in an *unc-104* mutant background (UNC-104 is required for UNC-17 transport to synapses), allowed us to precisely define the complete set of cells that stain for endogenous UNC-17 protein. We found the overlap of UNC-17 antibody staining with *cho-1* fosmid reporter expression to be the same as the overlap of *unc-17* fosmid reporter expression with *cho-1* fosmid reporter expression (*Figure 1F*), thereby validating the reliability of fosmid reporter expression patterns.

*unc-17*/VAChT expression defines cholinergic identity and is present in 52 of the 118 classes of adult hermaphroditic neurons, amounting to 159 out of 302 neurons (*Figure 1*, *Figure 2A*; *Table 1*;

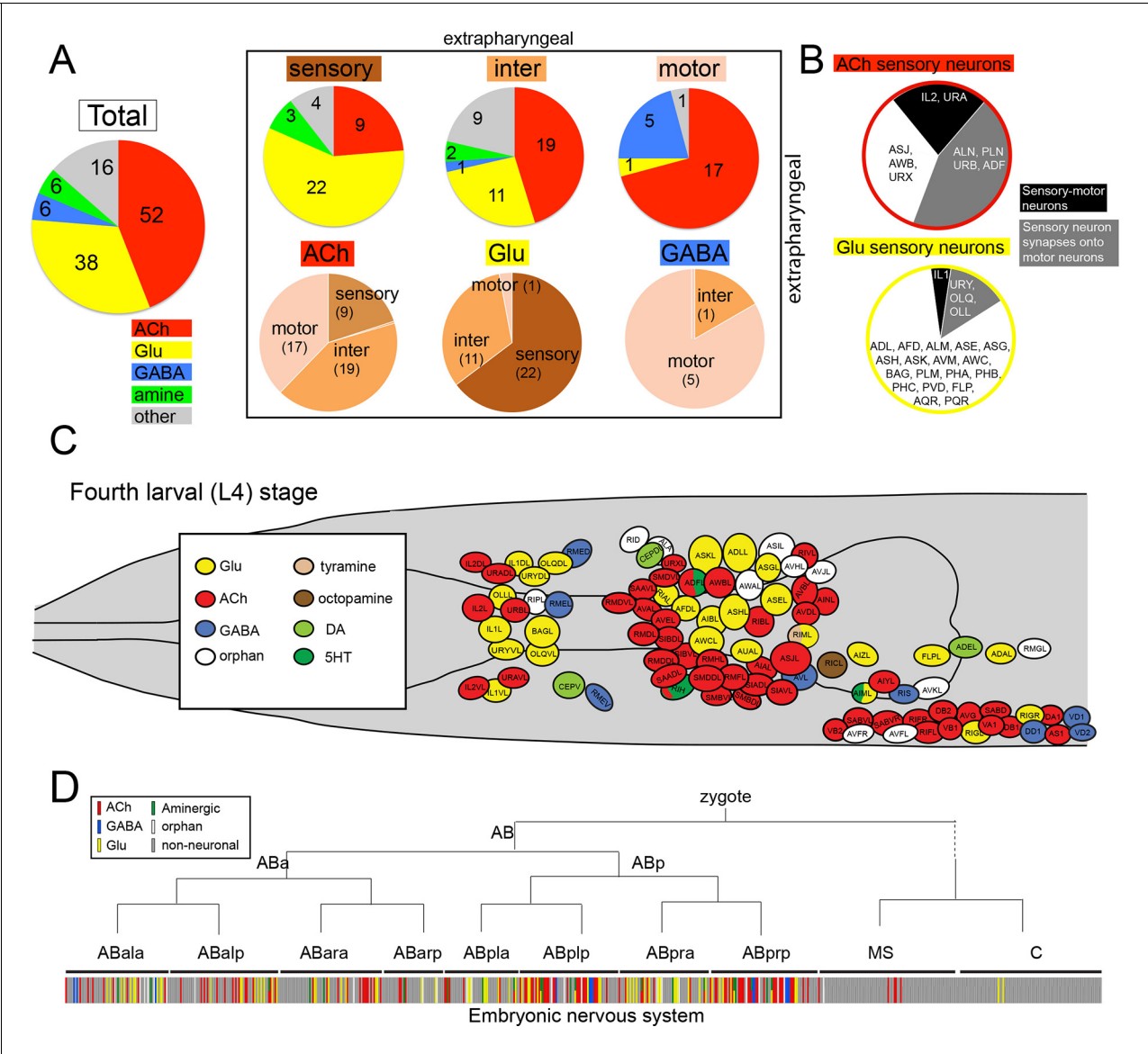

**Figure 2.** Distribution of neurotransmitters throughout the nervous system of the hermaphrodite. (**A**) Pie chart with numbers/distributions of cholinergic (this study), glutamatergic (*Serrano-Saiz et al., 2013*), GABAergic (*McIntire et al., 1993*) and aminergic (*Chase and Koelle, 2007*) neurons (including pharyngeal neurons). Inset: Pie charts of extrapharyngeal sensory, motor- and interneurons. Neurons that contain a classic fast transmitter plus an aminergic transmitter (e.g. RIH) are counted in the fast transmitter category. Classification of *C. elegans* neurons into sensory, inter- and motor neurons is complicated by the fact that a subset of sensory neurons are also motor neurons, i.e. synapse directly onto muscle (we count those neurons here only as sensory neurons). Conversely, a large number of motor neurons also extensively synapse onto other motor neurons or interneurons and hence classify as 'interneuron' as well; these neurons are shown exclusively in the motor neuron category. A number of neurons that were originally assigned as 'interneurons' by John White and colleagues are now considered motor neurons (because of the more recent identification of NMJs; e.g. SIA, SIB, SAB neurons), or are considered sensory neurons (because of their position in connectivity diagrams or expression of molecular markers; e.g. URA, URB, URXY, URY). See *Table 2* for a complete list of neurons and their neurotransmitter assignment. Lastly, we note that unpublished results from our lab demonstrate that at least two additional interneurons, not shown here, utilize GABA (M. Gendrel and O.H., unpubl. data). (**B**) Distance of sensory neurons to motor output (processing depth) of cholinergic and glutamatergic sensory neurons. (**C**) Location of neurons with different neurotransmitter identities in the head ganglia. (**D**) Neurotransmitter identity does not track with lineage history. Neurotransmitter identity is superimposed on the embryonic lineage diagram (*Sulston et al., 1983*), with each color line indicating one neuron type with a defined identity. White lines indicate no known neurotransmitter identity, gray lines indicate non-neuronal cells. Lines with two colors illustrate co-transmitter identities.

**Table 2.** Neurotransmitter map of the hermaphrodite nervous system.

| Neuron class | Neuron | Neurotransmitter | Notes |
|---|---|---|---|
| ADA | ADAL | Glu | |
| | ADAR | Glu | |
| ADE | ADEL | DA | |
| | ADER | DA | |
| ADF | ADFL | ACh & 5HT | |
| | ADFR | ACh & 5HT | |
| ADL | ADLL | Glu | |
| | ADLR | Glu | |
| AFD | AFDL | Glu | |
| | AFDR | Glu | |
| AIA | AIAL | ACh | |
| | AIAR | ACh | |
| AIB | AIBL | Glu | |
| | AIBR | Glu | |
| AIM | AIML | Glu & 5HT | |
| | AIMR | Glu & 5HT | |
| AIN | AINL | ACh | |
| | AINR | ACh | |
| AIY | AIYL | ACh | |
| | AIYR | ACh | |
| AIZ | AIZL | Glu | |
| | AIZR | Glu | |
| ALA | ALA | Unknown (orphan) | Newly assigned as mechanosensory (based on Sanders et al., 2013) |
| ALM | ALML | Glu | |
| | ALMR | Glu | |
| ALN | ALNL | ACh | Classified as sensory because of expression of oxygen sensors |
| | ALNR | ACh | |
| AQR | AQR | Glu | |
| AS | AS1 | ACh | |
| | AS2 | ACh | |
| | AS3 | ACh | |
| | AS4 | ACh | |
| | AS5 | ACh | |
| | AS6 | ACh | |
| | AS7 | ACh | |
| | AS8 | ACh | |
| | AS9 | ACh | |
| | AS10 | ACh | |
| | AS11 | ACh | |
| ASE | ASEL | Glu | |
| | ASER | Glu | |
| ASG | ASGL | Glu | |
| | ASGR | Glu | |

*Table 2 continued on next page*

Table 2 continued

| Neuron class | Neuron | Neurotransmitter | Notes |
|---|---|---|---|
| ASH | ASHL | Glu | |
| | ASHR | Glu | |
| ASI | ASIL | Unknown (orphan) | |
| | ASIR | Unknown (orphan) | |
| ASJ | ASJL | ACh | |
| | ASJR | ACh | |
| ASK | ASKL | Glu | |
| | ASKR | Glu | |
| AUA | AUAL | Glu | |
| | AUAR | Glu | |
| AVA | AVAL | ACh | |
| | AVAR | ACh | |
| AVB | AVBL | ACh | |
| | AVBR | ACh | |
| AVD | AVDL | ACh | |
| | AVDR | ACh | |
| AVE | AVEL | ACh | |
| | AVER | ACh | |
| AVF | AVFL | Unknown (orphan) | |
| | AVFR | Unknown (orphan) | |
| AVG | AVG | ACh | |
| AVH | AVHL | Unknown (orphan) | |
| | AVHR | Unknown (orphan) | |
| AVJ | AVJL | Unknown (orphan) | |
| | AVJR | Unknown (orphan) | |
| AVK | AVKL | Unknown (orphan) | |
| | AVKR | Unknown (orphan) | |
| AVL | AVL | GABA | |
| AVM | AVM | Glu | |
| AWA | AWAL | Unknown (orphan) | |
| | AWAR | Unknown (orphan) | |
| AWB | AWBL | ACh | |
| | AWBR | ACh | |
| AWC | AWCL | Glu | |
| | AWCR | Glu | |
| BAG | BAGL | Glu | |
| | BAGR | Glu | |
| BDU | BDUL | Unknown (orphan) | |
| | BDUR | Unknown (orphan) | |
| CAN | CANL | unknown MA (cat-1) | |
| | CANR | unknown MA (cat-1) | |
| CEP | CEPDL | DA | |
| | CEPDR | DA | |
| | CEPVL | DA | |

Table 2 continued on next page

*Table 2 continued*

| Neuron class | Neuron | Neurotransmitter | Notes |
|---|---|---|---|
| | CEPVR | DA | |
| DA | DA1 | ACh | |
| | DA2 | ACh | |
| | DA3 | ACh | |
| | DA4 | ACh | |
| | DA5 | ACh | |
| | DA6 | ACh | |
| | DA7 | ACh | |
| | DA8 | ACh | |
| | DA9 | ACh | |
| DB | DB1/3 | ACh | |
| | DB2 | ACh | |
| | DB3/1 | ACh | |
| | DB4 | ACh | |
| | DB5 | ACh | |
| | DB6 | ACh | |
| | DB7 | ACh | |
| DD | DD1 | GABA | |
| | DD2 | GABA | |
| | DD3 | GABA | |
| | DD4 | GABA | |
| | DD5 | GABA | |
| | DD6 | GABA | |
| DVA | DVA | ACh | |
| DVB | DVB | GABA | |
| DVC | DVC | Glu | |
| FLP | FLPL | Glu | |
| | FLPR | Glu | |
| HSN | HSNL | ACh & 5HT | |
| | HSNR | ACh & 5HT | |
| IL1 | IL1DL | Glu | Also a clear motor neuron |
| | IL1DR | Glu | |
| | IL1L | Glu | |
| | IL1R | Glu | |
| | IL1VL | Glu | |
| | IL1VR | Glu | |
| IL2 | IL2DL | ACh | Also a clear motor neuron |
| | IL2DR | ACh | |
| | IL2L | ACh | |
| | IL2R | ACh | |
| | IL2VL | ACh | |
| | IL2VR | ACh | |
| LUA | LUAL | Glu | |
| | LUAR | Glu | |

*Table 2 continued on next page*

*Table 2 continued*

| Neuron class | Neuron | Neurotransmitter | Notes |
|---|---|---|---|
| OLL | OLLL | Glu | |
| | OLLR | Glu | |
| OLQ | OLQDL | Glu | |
| | OLQDR | Glu | |
| | OLQVL | Glu | |
| | OLQVR | Glu | |
| PDA | PDA | ACh | |
| PDB | PDB | ACh | |
| PDE | PDEL | DA | |
| | PDER | DA | |
| PHA | PHAL | Glu | |
| | PHAR | Glu | |
| PHB | PHBL | Glu | |
| | PHBR | Glu | |
| PHC | PHCL | Glu | |
| | PHCR | Glu | |
| PLM | PLML | Glu | |
| | PLMR | Glu | |
| PLN | PLNL | ACh | |
| | PLNR | ACh | |
| PQR | PQR | Glu | |
| PVC | PVCL | ACh | |
| | PVCR | ACh | |
| PVD | PVDL | Glu | |
| | PVDR | Glu | |
| PVM | PVM | Unknown (orphan) | |
| PVN | PVNL | ACh | Only very few minor NMJs, more prominent neuron-neuron synapses |
| | PVNR | ACh | |
| PVP | PVPL | ACh | |
| | PVPR | ACh | |
| PVQ | PVQL | Glu | |
| | PVQR | Glu | |
| PVR | PVR | Glu | |
| PVT | PVT | Unknown (orphan) | |
| PVW | PVWL | Unknown (orphan) | |
| | PVWR | Unknown (orphan) | |
| RIA | RIAL | Glu | |
| | RIAR | Glu | |
| RIB | RIBL | ACh | |
| | RIBR | ACh | |
| RIC | RICL | Octopamine | |
| | RICR | Octopamine | |
| RID | RID | Unknown (orphan) | |

*Table 2 continued on next page*

Table 2 continued

| Neuron class | Neuron | Neurotransmitter | Notes |
|---|---|---|---|
| RIF | RIFL | ACh | |
| | RIFR | ACh | |
| RIG | RIGL | Glu | |
| | RIGR | Glu | |
| RIH | RIH | ACh & 5HT | |
| RIM | RIML | Glu & Tyramine | |
| | RIMR | Glu & Tyramine | |
| RIP | RIPL | Unknown (orphan) | |
| | RIPR | Unknown (orphan) | |
| RIR | RIR | ACh | |
| RIS | RIS | GABA | |
| RIV | RIVL | ACh | Only very few minor NMJs, more prominent neuron-neuron synapses |
| | RIVR | ACh | |
| RMD | RMDDL | ACh | |
| | RMDDR | ACh | |
| | RMDL | ACh | |
| | RMDR | ACh | |
| | RMDVL | ACh | |
| | RMDVR | ACh | |
| RME | RMED | GABA | |
| | RMEL | GABA | |
| | RMER | GABA | |
| | RMEV | GABA | |
| RMF | RMFL | ACh | |
| | RMFR | ACh | |
| RMG | RMGL | Unknown (orphan) | |
| | RMGR | Unknown (orphan) | |
| RMH | RMHL | ACh | |
| | RMHR | ACh | |
| SAA | SAADL | ACh | |
| | SAADR | ACh | |
| | SAAVL | ACh | |
| | SAAVR | ACh | |
| SAB | SABD | ACh | Makes clear neuromuscular junctions |
| | SABVL | ACh | |
| | SABVR | ACh | |
| SDQ | SDQL | ACh | |
| | SDQR | ACh | |
| SIA | SIADL | ACh | Makes clear neuromuscular junctions |
| | SIADR | ACh | |
| | SIAVL | ACh | |
| | SIAVR | ACh | |
| SIB | SIBDL | ACh | Makes clear neuromuscular junctions |

Table 2 continued on next page

*Table 2 continued*

| Neuron class | Neuron | Neurotransmitter | Notes |
|---|---|---|---|
| | SIBDR | ACh | |
| | SIBVL | ACh | |
| | SIBVR | ACh | |
| SMB | SMBDL | ACh | |
| | SMBDR | ACh | |
| | SMBVL | ACh | |
| | SMBVR | ACh | |
| SMD | SMDDL | ACh | |
| | SMDDR | ACh | |
| | SMDVL | ACh | |
| | SMDVR | ACh | |
| URA | URADL | ACh | Also a clear motor neuron |
| | URADR | ACh | |
| | URAVL | ACh | |
| | URAVR | ACh | |
| URB | URBL | ACh | |
| | URBR | ACh | |
| URX | URXL | ACh | |
| | URXR | ACh | |
| URY | URYDL | Glu | |
| | URYDR | Glu | |
| | URYVL | Glu | |
| | URYVR | Glu | |
| VA | VA1 | ACh | |
| | VA2 | ACh | |
| | VA3 | ACh | |
| | VA4 | ACh | |
| | VA5 | ACh | |
| | VA6 | ACh | |
| | VA7 | ACh | |
| | VA8 | ACh | |
| | VA9 | ACh | |
| | VA10 | ACh | |
| | VA11 | ACh | |
| | VA12 | ACh | |
| VB | VB1 | ACh | |
| | VB2 | ACh | |
| | VB3 | ACh | |
| | VB4 | ACh | |
| | VB5 | ACh | |
| | VB6 | ACh | |
| | VB7 | ACh | |
| | VB8 | ACh | |
| | VB9 | ACh | |

*Table 2 continued on next page*

*Table 2 continued*

| Neuron class | Neuron | Neurotransmitter | Notes |
|---|---|---|---|
| | VB10 | ACh | |
| | VB11 | ACh | |
| VC | VC1 | ACh | |
| | VC2 | ACh | |
| | VC3 | ACh | |
| | VC4 | ACh & 5HT | |
| | VC5 | ACh & 5HT | |
| | VC6 | ACh | |
| VD | VD1 | GABA | |
| | VD2 | GABA | |
| | VD3 | GABA | |
| | VD4 | GABA | |
| | VD5 | GABA | |
| | VD6 | GABA | |
| | VD7 | GABA | |
| | VD8 | GABA | |
| | VD9 | GABA | |
| | VD10 | GABA | |
| | VD11 | GABA | |
| | VD12 | GABA | |
| | VD13 | GABA | |
| **Summary for extrapharyngeal neurons** | | | |
| Sensory neuron: | | Sensory neuron: | |
| 38/104 classes | | ACh: 9 classes | |
| 87/282 total neurons | | Glu: 22 | |
| Motor neuron: | | GABA: 0 | |
| 24/104 | | Aminergic: 3 (all Dopa) | |
| 118/282 | | Unknown: 4 (ASI, AWA, PVM, ALA) | |
| Interneuron | | Motor neuron: | |
| 42/104 | | ACh: 17 classes | |
| 77/282 | | Glu: 1 (RIM) | |
| | | GABA: 5 | |
| | | Aminergic: 0 | |
| | | Unknown: 1 (RMG) | |
| | | Interneuron: | |
| | | ACh: 19 classes | |
| | | Glu: 11 | |
| | | GABA: 1 (RIS) | |
| | | Aminergic: 2 (CAN, RIC) | |
| | | Unknown: 9 | |
| **Pharyngeal neurons** | | | |
| I1 | I1L | ACh | Due to connectivity and rudimentary sensory endings, all polymodal |

*Table 2 continued on next page*

*Table 2 continued*

| Neuron class | Neuron | Neurotransmitter | Notes |
|---|---|---|---|
| | I1R | ACh | |
| I2 | I2L | Glu | |
| | I2R | Glu | |
| I3 | I3 | ACh | |
| I4 | I4 | Unknown (orphan) | |
| I5 | I5 | Glu & 5HT | |
| I6 | I6 | Unknown (orphan) | |
| M1 | M1 | ACh | |
| M2 | M2L | ACh | |
| | M2R | ACh | |
| M3 | M3L | Glu | |
| | M3R | Glu | |
| M4 | M4 | ACh | |
| M5 | M5 | ACh | |
| MC | MCL | ACh | |
| | MCR | ACh | |
| MI | MI | Glu | |
| NSM | NSML | 5HT | |
| | NSMR | 5HT | |

*Figure 1—figure supplement 2*). Compared to all other neurotransmitter systems, this makes ACh the most abundantly employed neurotransmitter system in *C. elegans* (Glu: 38 classes, GABA: 6 classes, Aminergic: 13 classes, six of which are exclusively aminergic; *Figure 2A*, *Table 2*). The abundance of ACh usage is illustrated in an even more striking manner if one considers the *C. elegans* connectome (*White et al., 1986*): 85% (100/118) of all neuron classes are innervated by a cholinergic neuron (*Table 3*). With one exception (the highly unusual CAN neurons, which show very little synaptic connectivity with any other neuron), all neurons that do not receive cholinergic input are either themselves cholinergic neurons or innervate neurons that are cholinergic (*Table 3*). In other words, all but one neuron class in the *C. elegans* nervous system are either cholinergic, receive cholinergic input or innervate a cholinergic neuron.

There does not appear to be any change in neurotransmitter identities in the first larval stage versus the adult stage, with the obvious exception of postembryonically generated neurons (mostly motor neurons). Expression of the cholinergic locus (*unc-17* and *cha-1*) commences in the 1.5-fold stage of embryogenesis and by the three-fold stage, expression is seen in all cholinergic neurons (data not shown).

*cho-1/*ChT expression extensively correlates with expression of *unc-17/*VAChT, both in terms of onset (by threefold stage; data not shown) and cellular specificity in the mature nervous system. In the hermaphrodite worm, all neurons that express *cho-1* also express *unc-17/VAChT* (even though expression of *unc-17* may be very low in at least one class, RIB), while 11 out of the 52 *unc-17(+)* classes do not express *cho-1* (half of these neuron classes are in the pharyngeal nervous system; *Figure 1*;*Table 1*; *Figure 1—figure supplement 2*). In contrast, as a previous analysis of small reporter gene fusions already suggested (*Combes et al., 2003*), expression of the acetylcholinesterase (AChE)-encoding *ace* genes does not correlate with *unc-17* expression. First, only one third of all cholinergic neuron classes express an *ace* gene (*Table 1*); and second, expression is observed in body wall muscle as well as in a few non-cholinergic neurons (*Figure 1H*). Given that the diffusible ACE proteins are secreted into the synaptic cleft, it may not come as a surprise that their site of synthesis does not necessarily match the site of ACh synthesis and release. The situation is similar in vertebrates; the only vertebrate AChE gene is expressed in cholinergic neurons, but the overlap is not complete and expression can also be

**Table 3.** Neurons receiving cholinergic inputs. Includes pharyngeal neurons. Data from www.wormwiring.org.

| Connectivity * | | Neuron class | # |
|---|---|---|---|
| Receiving ACh input | Cholinergic neurons | ADF, AIA, AIN, AIY, ALN, AS, ASJ, AVA, AVB, AVD, AVE, AWB, DA, DB, DVA, I3, IL2, M2, M4, PLN, PVC, PVN, PVP, RIB, RIF, RIH, RIR, RIV, RMD, RMF, RMH, SAA, SAB, SDQ, SIA, SIB, SMB, SMD, URA, URB, URX, VA, VB, VC | 44 |
| | Non-cholinergic neurons | ADA, ADE, ADL, AFD, AIB, AIM, AIZ, ALA, ALM, AQR, ASE, ASG, ASH, ASI, ASK, AUA, AVF, AVH, AVJ, AVK, AVL, ASA, AWC, BAG, BDU, CEP, DD, DVC, I2, I4, I5, IL1, LUA, M1, M3, MC, NSM, OLL, OLQ, PQR, PVQ, PVR, PVT, PVW, RIA, RIC, RID, RIG, RIM, RIP, RIR, RIS, RME, RMG, URY, VD | 56 |
| Receiving no ACh input | Cholinergic neuron | AVG, HSN, I1, M5, PDA, PDB | 6 |
| | Innervate cholinergic neuron | AVM, DVB, FLP, I6, MI, PDE, PHA, PHB, PHC, PLM, PVD, PVM | 12 |
| Neither of the above | | CAN | 1 |

observed in non-cholinergic neurons (*Gwyn and Flumerfelt, 1971*; *Levey et al., 1984*; *Reiss et al., 1996*).

A list of all *C. elegans* neurons with their presently assigned neurotransmitter identity is shown in *Table 2*. There are no overlaps in usage of the main neurotransmitter systems glutamate, GABA and ACh in the core nervous system of the hermaphrodite, but within the pharynx, one single neuron, the motor neuron M5, strongly expresses both cholinergic pathway genes *unc-17* and *cho-1* and, albeit very weakly, the glutamatergic marker *eat-4/VGLUT* (*Figure 1—figure supplement 2*). We visualized the general lack of overlap with a transgenic line that expresses three different, nuclear localized, fluorescent tags – one marking cholinergic neurons (*cho-1$^{fosmid}$::mChOpti)*, one marking glutamatergic neurons *(eat-4/VGLUT$^{fosmid}$::yfp)* and one marking all neurons (*rab-3$^{prom}$::bfp* strain) (*Figure 1G*; *Video 1*). There are, however, some overlaps of cholinergic identity with aminergic identity in the core nervous system: the ADF, HSN, RIH and VC4/5 neurons are cholinergic, but also serotonergic (*Duerr et al., 2001*; *Sze et al., 2000*); similarly, some glutamatergic neurons are also aminergic (*Serrano-Saiz et al., 2013*). The case of the postembryonically generated, hermaphrodite-specific VC motor neurons is particularly notable because of the distinct identities of specific VC subtypes. All six VC neurons express *unc-17/VAChT* and are therefore cholinergic, but VC4 and VC5 are also serotonergic (*Duerr et al., 1999*). Notably, the expression of serotonergic identity in VC4 and VC5 correlates with a failure to express *cho-1/*ChT, which is only expressed in VC1,2,3 and VC6 (*Figure 1—figure supplement 1*, *Figure 5C*). VC4 and VC5 innervate vulval muscles and some aspects of their identity (namely expression of the *unc-4* gene in VC4/5, but not VC1,2,3,6) are controlled by signals from vulval tissues (*Zheng et al., 2013*). We find that elimination of this vulval signal, or genetic elimination of the target muscle of the VC4/5 neurons (vulval muscle), does not impinge on the absence of *cho-1* expression in VC4/5 (data not shown).

## Relationship of cholinergic identity to neuron function, neuron position and neuronal lineage

ACh is used by sensory neurons, interneurons and motor neurons. Of the 45 extrapharyngeal cholinergic neuron classes, 9 are sensory neurons, 19 are interneurons and 17 are motor neurons (*Figure 2A*; *Table 2*; we only consider extrapharyngeal neurons because most pharyngeal neurons are polymodal, i.e. have sensory, inter- and motor neuron features; *Albertson and Thomson, 1976*; D.H.H. unpubl. data). Compared to other transmitter systems, motor neurons have a preference for employing ACh (17/24 extrapharyngeal motor neuron classes are cholinergic; *Figure 2A*; *Table 2*). In contrast, sensory neurons are predominantly glutamatergic (22/38 use Glu), but there is nevertheless an appreciable number of cholinergic sensory neurons (9/38 extrapharyngeal sensory neuron classes use ACh; *Table 2*). Intriguingly, most cholinergic sensory neurons have very shallow processing depth, i.e. are closely connected to the motor system (*Figure 2B*). Two (IL2 and URA) directly synapse onto muscle (i.e. are sensory-motor neurons), another four (ALN, PLN, ADF and URB) synapse directly onto motor neurons, while another two (URX and AWB) synapse onto cholinergic command interneurons that innervate motor neurons. The latter two cases are the only cases in the entire *C. elegans* nervous system where a multi-neuron pathway from sensory, via inter- to motor

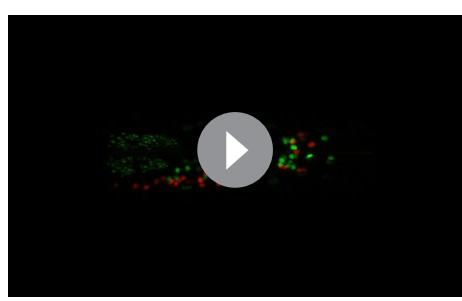

**Video 1.** Cholinergic and glutamatergic head neurons. Confocal image stack of a transgenic worm expressing *cho-1::mChopti (otIs544)* and *eat-4::yfp (otIs388) fosmid* reporter gene constructs in the head.

neurons is entirely made of exclusively one neurotransmitter system. In comparison, glutamatergic sensory neurons do not display such a narrow processing depth (*Figure 2B*).

The predominance of ACh as a neurotransmitter does not solely stem from its widespread usage in motor neurons. ACh is also the most broadly used neurotransmitter of interneurons (19 classes; compared to 11 glutamatergic; *Figure 2A*; *Table 1*). In comparison to ACh and glutamate (Glu), the neurotransmitter GABA is only very sparsely used by interneurons (presently only 1 class; [*McIntire et al., 1993*]); this is no reflection of a paucity of inhibitory neurotransmission in *C. elegans* since both ACh and Glu can act as inhibitory neurotransmitter through the gating of postsynaptic chloride channels (see below).

The most notable set of interneurons to which we assigned a cholinergic neurotransmitter identity are the command interneurons, which are well-characterized central integrators of information flow in the nervous system that directly synapse onto motor neurons (*Chalfie et al., 1985*; *Von Stetina et al., 2006*). Their neurotransmitter identity was previously not known and we verified their cholinergic identity through a number of different co-stains (summarized in *Figure 1—figure supplement 1*; see Materials and methods). Expression of the *cho-1* fosmid reporter overlapped with expression of the glutamate receptors *glr-1* and *nmr-1* in the AVA, AVE, AVD and PVC command interneurons. To confirm the cholinergic identity of the AVB command interneuron, we crossed the *cho-1* fosmid reporter with *sra-11* and *acr-15* reporters. Overlap of *cho-1* with these two reporters allowed us to confirm that AVB expresses cholinergic identity genes. All command interneurons showed expression of the *unc-17* fosmid reporter.

Apart from assigning cholinergic neurotransmitter identity to different types of neurons (sensory vs. inter vs. motor neurons), we examined whether cholinergic neurotransmitter identity correlates with other intrinsic neuronal features. We find that the adoption of cholinergic neurotransmitter identity does not correlate with position of the neuron within the nervous system, as shown in *Figure 2C*, with the notable exception of cholinergic motor neurons in the ventral head ganglion and along the ventral nerve cord. There is no correlation between the adoption of cholinergic identity and developmental history of the neurons. We arrived at this conclusion by mapping neurotransmitter identity onto the entire lineage diagram and not detecting any obvious lineage clusters of cells that uniquely employ ACh (or any other neurotransmitter; *Figure 2D*).

## Distribution of cholinergic neurons, as well as other neurotransmitters, in relation to the entire connectome

With the identification of the complete set of cholinergic neurons, and with the consideration of previously identified glutamatergic, GABAergic and monoaminergic neurons, a neurotransmitter identity can now be assigned to ~90% of all neuron classes (102/118) and total neurons (275/302; *Table 2*). While some of the remaining orphan neurons (e.g. the prominent olfactory neuron AWA) contain small synaptic vesicles that are indicative of the usage of an as yet uncharacterized neurotransmitter system, about half of the remaining 16 'orphan' neuron classes display, according to John White's EM analysis, a notable paucity or even absence of synaptic vesicles and/or are predominated by dark staining vesicles (e.g. AVF, AVH, AVJ, RIP) (*White et al., 1986*), suggesting that these neurons either signal mostly via electrical synapses or via neuropeptides.

The assignment of neurotransmitter identity to ~90% of neurons prompted us to take a system-wide view of neurotransmitter usage. We started by examining neurotransmitter usage within a number of specific circuitries described by John White and colleagues, including circuitries associated with amphid sensory neurons, with head motor neurons and with motor neurons in the ventral nerve cord (*White et al., 1986*). While some circuitries show a mixed usage of different neurotransmitter systems (one example shown in *Figure 3A*), the circuitry associated with the motor neurons of the

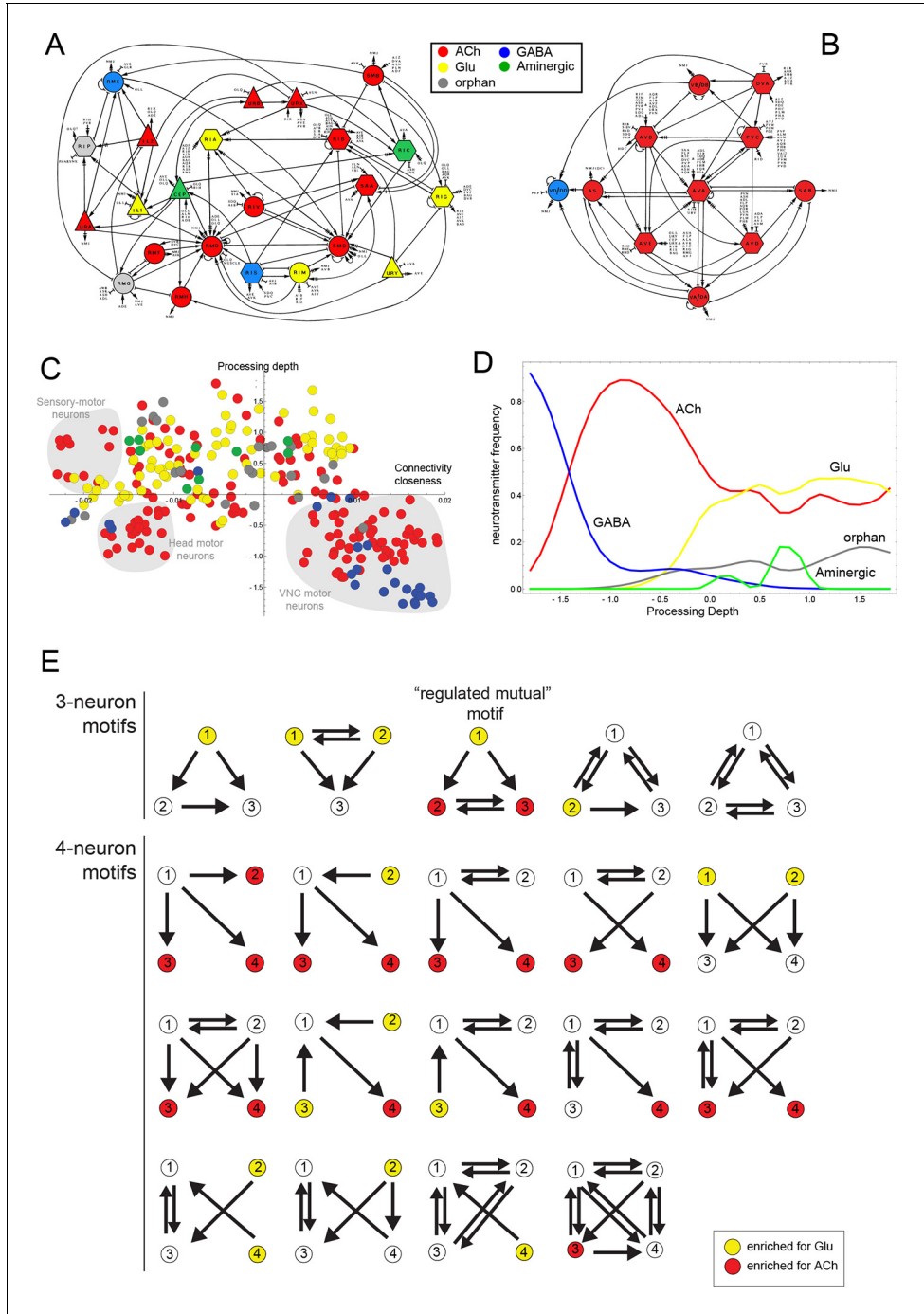

**Figure 3.** Neurotransmitter distribution in nervous system-wide circuit diagrams. (**A, B**) Circuit diagrams, taken from *White et al. (1986)*, with neurotransmitter identities added in colors, as indicated. Panel A shows what White et al. called the "Circuitry associated with motoneurons in the nerve ring" and panel B shows the "Circuitry associated with the motoneurons of the ventral cord". (**C**) A visualization of the *C. elegans* connectome that reflects signal flow through the network as well as the closeness of neurons in the network, as previously proposed and described (*Varshney et al., 2011*). Coordinates from the diagram were kindly provided by Lav Varshney. The vertical axis represents the signal flow depth of the network, i.e. the number of synapses from sensory to motor neurons. The horizontal axis represents connectivity closeness. We superimpose here neurotransmitter identity onto this network diagram, illustrating some network cluster enriched for ACh usage (shaded gray). (**D**) A graphic representation that focuses on processing depth, illustrating whether a neurotransmitter is used more frequently in upper (sensory) or lower (motor) layers of the network. (**E**) Network motifs enriched in the *C. elegans* connectome and their neurotransmitter usage. Colors indicate if the neurons in this position are enriched for the usage of Glu or ACh.

ventral nerve cord show the striking feature of being mainly cholinergic (with the exception of the GABAergic DD/VD motor neurons; *Figure 3B*). That is, not only do most motor neurons ('MNs') (SAB, DA, DB, VA, VB, AS) employ ACh, but all neurons that innervate these neurons (and that are also strongly interconnected among each other) are also cholinergic. This includes the command interneurons AVA, AVB, AVE, AVD, PVC, as well as the DVA interneuron, which is also closely associated with the motor circuit (*Figure 3B*). Due to the extent of their interconnectedness, this group of six interneuron classes has previously been defined as a 'rich club' of neurons (*Towlson et al., 2013*). The adoption of cholinergic identity within an entire functional circuit prompts the immediate question whether activity of the circuit plays a role in the expression of cholinergic genes. However, we find that genetic silencing of the *C. elegans* nervous system, achieved through elimination of the *snb-1/synaptobrevin* gene, has no impact on the expression of cholinergic identity markers in arrested L1 larvae (data not shown).

Taking a broader view we mapped neurotransmitter identity on a wiring diagram that reflects signal flow through the network as well as connectivity closeness of neurons in the network, as suggested by Varshney and colleagues (*Varshney et al., 2011*) (*Figure 3C*). We also examined the parameter of 'processing depth' in isolation, as had been done previously (*Varshney et al., 2011*). We considered the distance of each neuron from sensory input to motor output, assigned this relative position the parameter 'processing depth' and then the portion of neurons that use each neurotransmitter at each processing depth (*Figure 3D*). Both types of representations quantify and effectively visualize what the identity of many of the cholinergic neurons already suggested: compared to other neurotransmitter systems (particularly Glu) ACh is enriched, but not exclusively located to lower levels of information processing. Another notable feature of this presentation is that it visualizes the connectivity closeness of distinct clusters of cholinergic motor neurons (shaded in gray in *Figure 3C*); these neurons are the head sensory-motor neurons, head motor neurons and the above-mentioned ventral nerve cord (VNC) motor neuron circuitry.

We considered neurotransmitter usage within the several types of recurring network motifs, composed of three or four neurons, which have been described to be enriched within the *C. elegans* connectome, such as feedforward motifs of three neurons (*Milo et al., 2002*). Our goal was to examine whether the usage of ACh (or any other neurotransmitter system) is biased for certain positions of a neuron within these motifs. Using previously described approaches (*Milo et al., 2002*) (see Materials and methods), we identified five 3-neuron motifs and fourteen 4-neuron motifs that are significantly enriched in the *C. elegans* connectome using the latest connectivity dataset (*Figure 3E*). We found either ACh or Glu to be enriched in specific positions in all but one of these motifs. ACh was enriched at a specific position in 11 out of these 19 motifs. Generally, there is a strong trend of ACh being more frequently used at the downstream end of the signaling flow within specific motifs, while Glu tends to be located at upstream positions within motifs (*Figure 3E*), which is consistent with the processing depth analysis described above (*Figure 3D*).

In one 3-neuron motif, previously termed a 'regulated mutual motif' (*Milo et al., 2004*), each one of the interconnected neurons is enriched for a specific neurotransmitter and we examined this motif more closely for reasons that will become evident in later sections of this paper. The general architecture of this motif is defined by one neuron ('A') innervating two reciprocally connected neurons ('B' and 'C'; *Figure 3E*). 224 occurrences of this motif can be found in the *C. elegans* hermaphroditic connectome. This motif is significantly enriched for the presence of cholinergic neurons in either position 2 or 3, or in both. Notably, position 1 is significantly enriched for Glu usage. In 146 out of the 224 motif occurrences, ACh is used by both neuron '2' and '3' (listed in *Table 4*), and 134 of these 146 motifs break down into a number of two striking types. In the first type, reciprocally connected command interneurons are either innervated by a sensory neuron or by an interneuron (*Table 4*). In virtually all of these cases, the innervating sensory neurons are glutamatergic. In many cases, the reciprocally connected command interneurons are neurons that control different directions of movement (forward vs. reverse; *Table 4*). In the second type, the SMD or RMD head motor neurons are reciprocally connected and innervated again either by mostly glutamatergic sensory neurons or by interneurons (*Table 4*).

## Mapping putative inhibitory cholinergic synapses

The *C. elegans* genome encodes not only conventional, excitatory ACh-gated cation channels, but also inhibitory ACh-gated anion channels (*Hobert, 2013*; *Putrenko et al., 2005*). Based on the

**Table 4.** Occurences of the 'Regulated Mutual' network motif.

| Type 1: Sensory>command interneurons | | | | Type 2: Interneurons>command interneurons | | | | Type 3: Sensory neurons>head motor neurons | | | Type 4: Interneurons>head motor neurons | | | Type 5: Egg laying circuit | | | Miscellaneous | | |
|---|---|---|---|---|---|---|---|---|---|---|---|---|---|---|---|---|---|---|---|
| SN | CI | CI | opp. | IN | CI | CI | opp. | SN | hMN | hMN | IN | hMN | hMN | | | | | | |
| ADEL | AVAL | AVAR | | ADAL | AVAL | AVAR | | CEPVL | RMDDL | RMDVR | RIAL | RMDDL | RMDVR | AIML | AVFL | AVFR | PHBL | VA12 | AVAL |
| ADER | AVAL | AVAR | | ADAL | AVAR | AVBL | yes | IL1DL | RMDDR | RMDVL | RIAL | RMDDR | RMDVL | AIMR | RIFR | HSNR | PHCL | VA12 | AVAL |
| ADER | AVAR | AVDR | | ADAL | AVAR | AVBR | yes | IL1DR | RMDDL | RMDVR | RIAR | RMDDL | RMDVR | AIMR | AVFL | AVFR | VA12 | PVCL | PVCR |
| ADER | AVAR | AVER | | ADAR | AVAR | AVBL | yes | IL1L | RMDDL | RMDVR | RIAR | RMDDR | RMDVL | AIMR | AVFL | HSNR | AVEL | DA01 | AS01 |
| ADLL | AVAL | AVDL | yes | ADAR | AVAR | AVBR | yes | IL1L | RMDL | RMDR | RICR | SMDDL | SMDVR | AIMR | AVFR | HSNL | AVER | DA01 | AS01 |
| ADLL | AVAL | AVBL | | ADAR | AVAR | AVDR | | IL1R | RMDDR | RMDVL | RICR | SMDDR | SMDVL | HSNL | AVFL | HSNR | AVHL | ADFR | AWBR |
| ADLL | AVAR | AVDR | | ALA | AVAR | AVER | yes | IL1R | RMDL | RMDR | RIML | SMDDR | SMDVL | | | | AWAR | ADFR | AWBR |
| ADLL | AVAR | AVBL | | AUAR | AVAR | AVER | yes | IL1VL | RMDDL | RMDVR | RIMR | SMDDL | RMDR | | | | CEPVR | IL2VR | URAVR |
| ADLL | AVAR | AVDR | | AVBR | AVAL | AVDL | | IL1VR | RMDDR | RMDVL | RIMR | SMDDL | SMDVR | | | | | | |
| ADLR | AVAR | AVBL | | AVDL | AVAR | AVDR | | IL2L | RMDL | RMDR | RIS | RMDL | RMDR | | | | | | |
| ADLR | PVCL | PVCR | | AVEL | AVAR | AVAR | yes | OLLL | SMDDL | SMDVR | RIVR | SMDDL | SMDVR | | | | | | |
| ADLR | AVAL | AVAR | | AVER | AVAR | AVDL | yes | OLLR | SMDDL | SMDVR | RMGR | RMDL | RMDR | | | | | | |
| ADLR | AVAL | AVDL | | AVER | PVCL | AVEL | yes | URYDL | RMDDR | RMDVL | | | | | | | | | |
| ALML | AVAR | AVBL | | AVG | AVAL | AVAR | | URYDR | RMDDL | RMDVR | unc-42 | unc-42 | unc-42 | | | | | | |
| AQR | AVAR | PVCR | yes | AVG | AVAL | AVBL | | URYVL | RMDDL | SMDVR | | | | | | | | | |
| AQR | AVAR | AVBL | yes | AVG | AVAR | AVBR | yes | URVVR | RMDDR | RMDVL | | | | | | | | | |
| AQR | AVAL | AVER | | AVG | AVAR | AVDR | yes | | | | | | | | | | | | |
| AQR | AVAR | PVCR | | AVJR | AVAR | AVBL | yes | | | | | | | | | | | | |
| AQR | AVAR | AVDL | yes | AVJR | AVAR | AVDR | | | | | | | | | | | | | |
| ASHL | AVAL | AVDL | yes | AVJR | AVAR | AVER | | | | | | | | | | | | | |
| ASHR | AVAR | AVBR | yes | AVJR | AVAL | PVCL | | | | | | | | | | | | | |
| ASHR | AVAR | AVER | | AVJR | AVAR | PVCR | yes | | | | | | | | | | | | |
| ASHR | AVAR | PVCL | | BDUR | PVCL | PVCR | | | | | | | | | | | | | |
| AVM | PVCL | PVCR | | DVA | AVAL | AVAR | | | | | | | | | | | | | |
| BAGL | AVAR | AVER | | DVC | AVAL | AVDL | | | | | | | | | | | | | |
| FLPL | AVAL | AVAR | yes | LUAR | AVAL | AVDL | | | | | | | | | | | | | |
| FLPL | AVAL | AVDL | yes | LUAR | AVAL | PVCR | yes | | | | | | | | | | | | |
| FLPL | AVAL | PVCR | | PVCR | AVDL | AVEL | yes | | | | | | | | | | | | |
| FLPL | AVAR | AVBL | | PVNL | AVAL | AVDL | yes | | | | | | | | | | | | |
| FLPL | AVAR | AVBR | yes | PVNL | AVAL | PVCL | | | | | | | | | | | | | |
| FLPL | AVAR | AVDR | yes | PVPL | AVAL | AVAR | yes | | | | | | | | | | | | |
| FLPL | AVAR | PVCR | | PVPL | AVAL | PVCL | yes | | | | | | | | | | | | |
| FLPR | AVAL | AVAR | yes | | | | | | | | | | | | | | | | |

*Table 4 continued on next page*

*Table 4 continued*

| Type 1: Sensory neurons>command interneurons | | | opp. | Type 2: Interneurons>command interneurons | | | opp. | Type 3: Sensory neurons>head motor neurons | | Type 4: Interneurons>head motor neurons | Type 5: Egg laying circuit | Miscellaneous |
|---|---|---|---|---|---|---|---|---|---|---|---|---|
| FLPR | AVAL | AVDL | | PVPL | AVAL | PVCR | yes | | | | | |
| FLPR | AVAR | AVBL | yes | PVPL | AVAR | AVBL | yes | | | | | |
| FLPR | AVAR | AVBR | yes | PVPL | AVAR | AVBR | yes | | | | | |
| FLPR | AVAR | AVDR | | PVPL | AVAR | AVDR | | | | | | |
| FLPR | AVAR | AVER | | PVPL | AVAR | PVCL | yes | | | | | |
| FLPR | AVDL | AVEL | | PVPL | AVAR | PVCR | yes | | | | | |
| PHBL | AVAL | AVAR | | PVPL | PVCL | PVCR | | | | | | |
| PHBL | AVAL | AVDL | | PVPR | AVAR | AVBR | yes | | | | | |
| PHBL | AVAL | PVCL | yes | PVPR | AVAR | PVCL | yes | | | | | |
| PHBL | AVAR | PVCL | yes | PVPR | AVAR | PVCR | yes | | | | | |
| PHBR | AVAL | AVAR | | PVPR | PVCL | PVCR | | | | | | |
| PHBR | AVAL | AVDL | | RIBR | AVAR | AVER | | | | | | |
| PHBR | AVAL | PVCL | yes | RICL | AVAL | AVAR | | | | | | |
| PHBR | AVAR | PVCR | | RICR | AVAL | AVAR | | | | | | |
| PHBR | AVAR | PVCL | yes | SDQL | AVAL | AVAR | | | | | | |
| PHBR | AVAR | PVCR | | SDQL | AVAL | AVDL | | | | | | |
| PHBR | PVCL | PVCR | yes | | | | | | | | | |
| PHCL | AVAL | PVCL | yes | | | | | | | | | |
| PQR | AVAL | AVAR | | | | | | | | | | |
| PQR | AVAL | AVDL | | | | | | | | | | |
| *unc-3* | *unc-3* | | | *unc-3* | *unc-3* | | | | | | | |

Yellow: Glu, red: ACh, green: Aminergic, blue: GABA.

opp.: command interneurons control opposite drives (forward/reverse).

SN: sensory neuron, IN: interneuron, CI: command interneuron, hMN: head motor neuron. Black bar: Transcription factor controlling cholinergic identity. Note that most interconnected neurons are controlled by the same transcription factor.

synaptic connectivity diagram and the knowledge of the identity of all cholinergic neurons, it is therefore possible to predict potential inhibitory cholinergic transmission by examining which neurons express an ACh-gated anion channel. The *C. elegans* genome encodes at least four ACh-gated anion channels, *acc-1* through *acc-4,* two of which (*acc-1* and *acc-2*) were electrophysiologically validated to be inhibitory receptors, while the function of two others (*acc-3* and *acc-4*) is less clear (**Putrenko et al., 2005**). We examined their expression pattern using available but previously uncharacterized fosmid-based reporter constructs (**Sarov et al., 2012**).

An *acc-3* fosmid reporter showed no appreciable expression throughout the animal, whereas *acc-1* and *acc-2* fosmid reporters show very restricted and non-overlapping expression in the adult nervous system (**Figure 4**). The *acc-1* fosmid reporter is expressed in a subset of cholinergic neurons, including cholinergic neurons in the ventral nerve cord, the retrovesicular ganglion and a few head neurons (including the SMD, RMD motor neurons, the AVA and AVE command interneurons and the SAA neurons). A small number of glutamatergic neurons also express *acc-1* (including the pharyngeal neurons MI and M3, the PLM neurons and an unidentified neuronal pair in the lateral ganglion). The *acc-2* fosmid reporter is expressed in a distinct, small set of glutamatergic neurons (RIA, RIG, PHA, AIZ) and cholinergic neurons (URX, RIH). We also found that the *acc-2* fosmid reporter is strongly expressed in the newly identified male-specific MCM neurons.

The *acc-4* fosmid reporter showed the most striking expression pattern. As assessed by coexpression with *cho-1,* the *acc-4* fosmid reporter is expressed almost exclusively in almost all of the 52 classes of cholinergic neurons (**Figure 4**). The only cholinergic neuron classes not expressing *acc-4* are

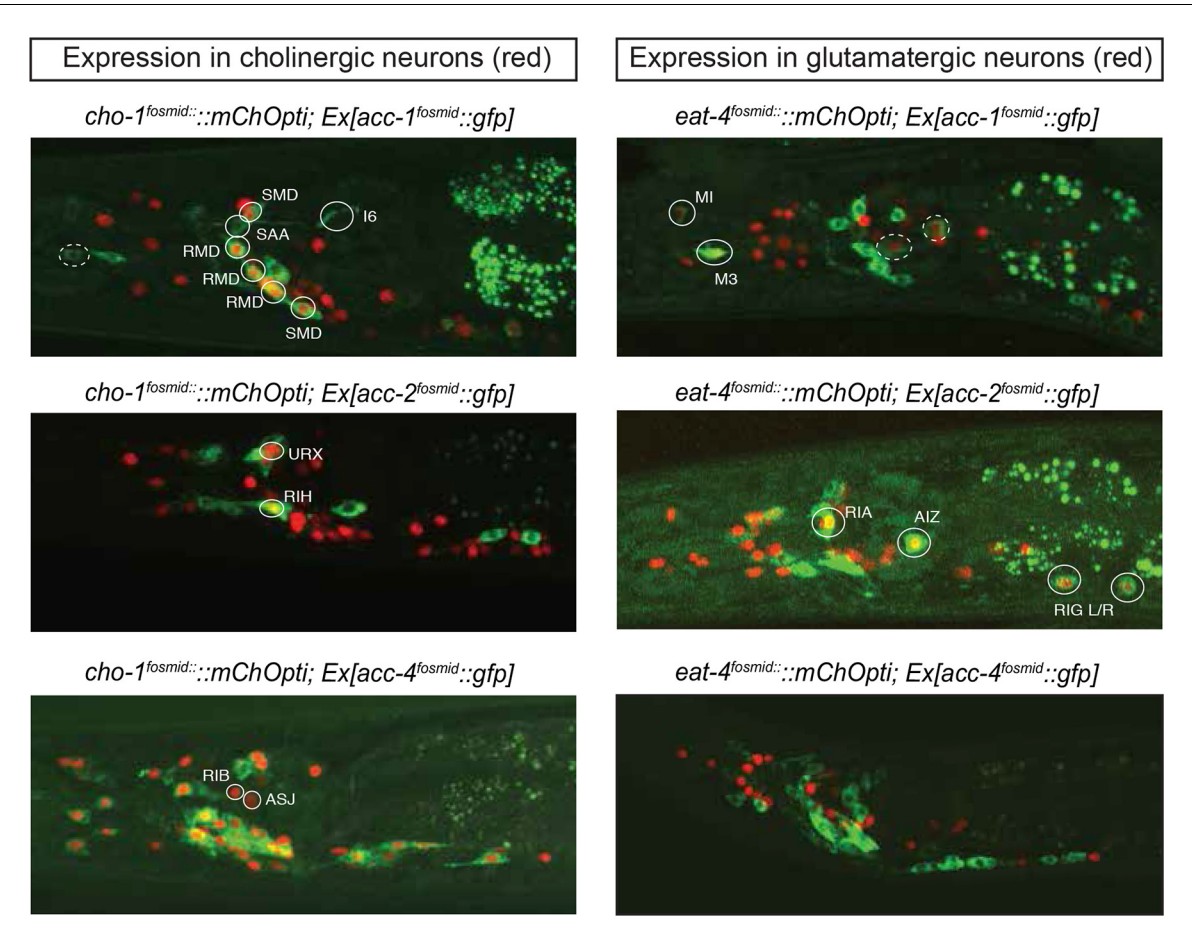

**Figure 4.** Expression pattern of ACh-gated chloride channels. Expression pattern of *acc* fosmid reporters in L4 stage animals are shown. Transgenes: *otEx6374* = *acc-1* fosmid reporter; *otEx6375* = *acc-2* fosmid reporter; *otEx6376* = *acc-4* fosmid reporter; *otIs545* = *cho-1* fosmid reporter; *otIs518* = *eat-4* fosmid reporter. Besides the neurons shown here, *acc-1* and *acc-2* are expressed in a small number of additional neurons (not shown).

the ASJ and RIB neurons and the only *acc-4*-expressing non-cholinergic neurons are the AVF neurons. If *acc-4* indeed is able to operate as inhibitory receptor (as suggested by its sequence), this expression data indicates that most cholinergic neurons can be silenced by presynaptically released ACh. In line with this prediction, more than half of all cholinergic neurons are innervated by cholinergic neurons. Among those neurons are the cholinergic command interneurons. This is particularly intriguing in light of laser ablation, electrophysiological and modeling data which indicate that specific command interneuron classes inhibit each others activity (*Rakowski et al., 2013*; *Roberts et al., 2016*). Another notable case of likely cross-inhibitory cholinergic connection is between members of two distinct head motor neuron classes (RMD and SMD classes). Notably, both the cross-inhibitory command interneurons and cross-inhibitory head motor neurons are parts of the above-described 'regulated mutual' network motif in which inter-connected cholinergic neurons are innervated by the same upstream neuron (*Figure 3E*). Regulated mutual motifs with negative interactions can operate as toggle switches that commit to one specific drive (forward movement) while inhibiting the alternative (reversal) drive.

However, it is important to keep in mind that a number of cholinergic neurons (including the command interneurons, but also VNC MNs) are also known to express excitatory ACh-gated ion channels (*acr* genes; www.wormbase.org), indicating that cholinergic input into these neurons may be complex.

We furthermore note that a substantial number of cholinergic neurons that express *acc-4* are not innervated by cholinergic neurons (as predicted by the connectome), raising the intriguing possibility that ACC-4 may act as an inhibitory autoreceptor on cholinergic neurons. In the context of gene expression networks, negative autoregulation can confer a number of useful functions, including speed-up of circuit responses and noise reduction (*Hart and Alon, 2013*). A substantiation of this hypothesis will require a determination of the localization of ACC-4 protein as well as additional subunits with which ACC-4 must act to constitute an inhibitory receptor (*Putrenko et al., 2005*), a feat beyond the scope of this present study.

## Acetylcholine is also the most broadly used neurotransmitter in the male-specific nervous system

The *C. elegans* male contains 91 sex-specific neurons, defining 24 classes, most of them located in the tail. We find that 16 out of these 24 classes are cholinergic (*Figure 5A*, *Figure 1—figure supplement 3*, *Table 5*). These cholinergic neurons include the only male-specific head neurons (the CEM sensory neurons) and an additional, male-specific class of motor neurons in the ventral nerve cord, the CA neurons. The three key themes observed in the hermaphrodite nervous system also apply to the male-specific neurons: (1) ACh is the most broadly used neurotransmitter in the male nervous system; (2) ACh is used in sensory, inter- and motor neurons of the male-specific nervous system; (3) the male-specific sensory neurons that are cholinergic are all in close proximity to the motor circuitry: most of them directly innervate muscle (i.e. are sensory/motor neurons; PCB, PCC, SPC; several ray neurons) while all others (HOB, SPV) innervate motor neurons. Like in the pharyngeal nervous system, we found neurons labeled by two conventional fast transmitters markers– the PVV neurons and the R6A neurons express *unc-17/VAChT* and *eat-4/VGLUT* (data not shown). ACh/Glu cotransmission has been observed in some central synapses in the vertebrate central nervous system (*Nishimaru et al., 2005*; *Ren et al., 2011*).

Most of the male-specific neurons are generated postembryonically from embryonically generated blast cells that divide during larval stages. One notable exception is the male-specific head sensory neuron class CEM. The two pairs of CEM neurons are generated in the embryo in both sexes, but are removed specifically in the hermaphrodite through programmed cell death (*Sulston et al., 1983*). We examined the onset of cholinergic differentiation of these neurons in males and found that they only start expressing cholinergic identity features at the L4 larval stage (*Figure 5A*). Hence, even though generated in the embryo, long before sexual maturation, neurotransmitter identity of CEM male-specific neurons only becomes established during overt sexual maturation in late larval stages. The same applies to the two classes of hermaphrodite-specific cholinergic neurons, the HSN and VC neurons. HSN is born embryonically, and VCs are born in the first larval stage, yet onset of cholinergic pathway genes is only observed in late L4 larval stages (*Figure 5B–C*). The late onset of neurotransmitter expression in the VC neurons is particularly notable if one compares the onset of cholinergic marker expression in the VC neurons with other cholinergic motor neurons born at the

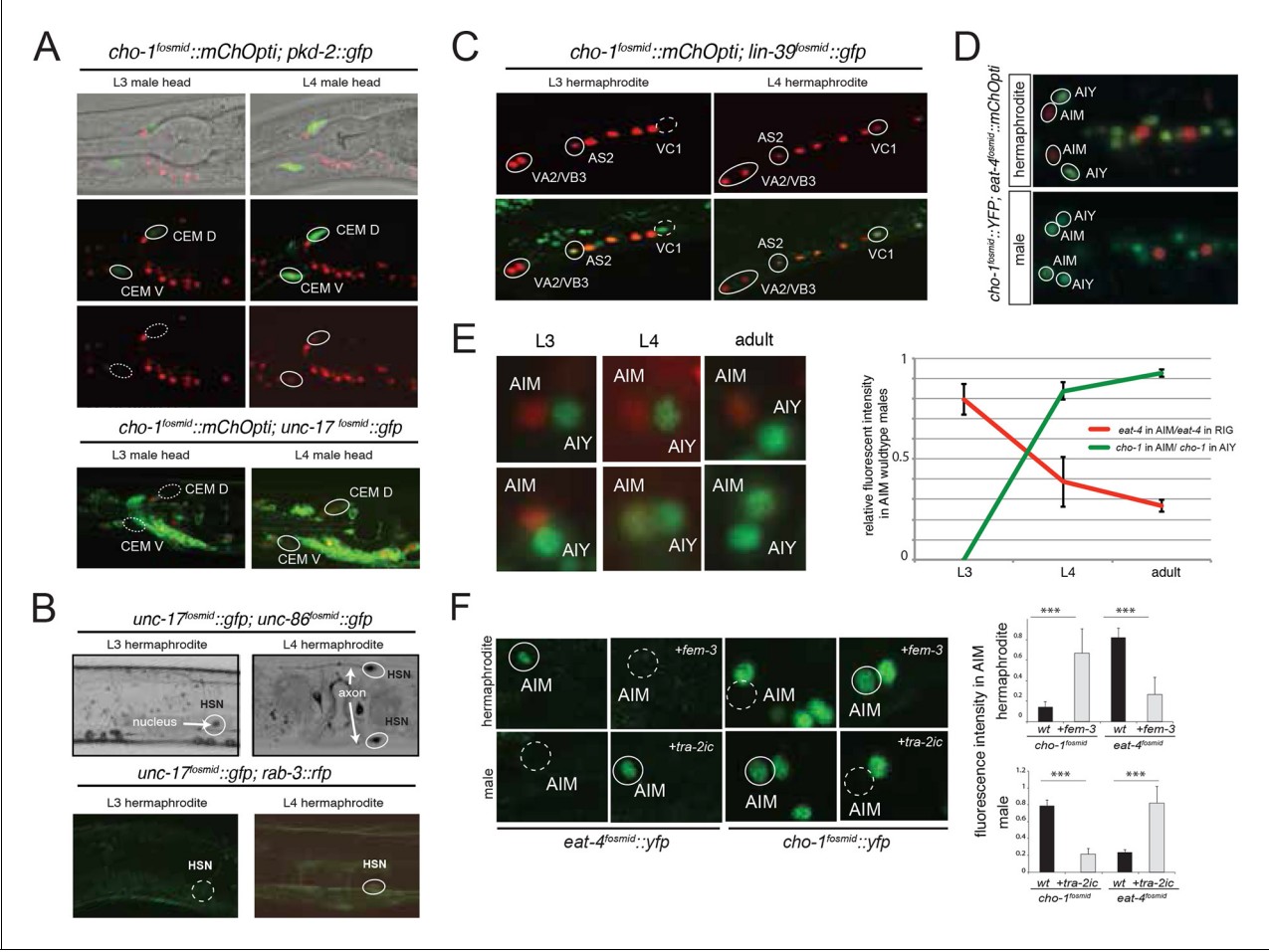

**Figure 5.** Sexual and temporal dynamics of cholinergic identity. (**A**) Male-specific CEM neurons are cholinergic, but turn on *cho-1* (*otIs544*) and *unc-17* (*otIs576*) only in late L4. In the top panels CEM neurons are labeled by the *pkd-2* reporter (*bxIs14*). See *Figure 1—figure supplement 3* and *Table 5* for a list of all male-specific cholinergic neurons. (**B**) Hermaphrodite-specific HSN neurons turn on the cholinergic marker *unc-17* and pan-neuronal *rab-3* also in late L4. HSN neurons are labeled by a nuclear localized *unc-86* fosmid reporter (*otIs337*). At L4 and later stages, *unc-17* fosmid expression (*otIs576*) becomes apparent in both soma and axon (top panels). The expression of the pan-neuronal marker *rab-3* (*otIs355*) is also first observed in late L4 (bottom panels). (**C**) Hermaphrodite-specific VC neurons turn on *unc-17* and *cho-1* only in late L4 (note that *cho-1* is NOT in VC4/5); this is later than the onset of the same genes in VA and VB neurons (VA, VB and VC neurons are labeled with the HOX gene *lin-39*). Transgenes: *wgIs18* = *lin-39* fosmid reporter; *otIs544* = *cho-1* fosmid reporter. (**D**) Sexually dimorphic neurotransmitter identity of a sex-shared neuron class. The AIM neuron expresses *cho-1* (and *unc-17*; not shown) in adult males, but expresses *eat-4/VGLUT* instead in hermaphrodites Transgenes: *otIs354* = *cho-1* fosmid reporter; *otIs518* = *eat-4* fosmid reporter. (**E**) Sexually dimorphic neurotransmitter switch. Until the L3 stage, both male and hermaphrodite AIM neurons are glutamatergic (express *eat-4/VGLUT*). While hermaphrodites continue to express *eat-4*, males downregulate *eat-4* and turn on *cho-1* (and *unc-17*; not shown). (**F**) The neurotransmitter switch is cell-autonomously controlled by the sex-determination pathway. In the upper panels, the masculinizing *fem-3* gene is force-expressed in the AIM neurons (with the *eat-4prom11* driver) in otherwise hermaphroditic animals; in the lower panels, the masculinizing *tra-2* intracellular domain ('*tra-2ic*') is expressed in AIM neurons of the male. Quantification is provided on the right.

same time, namely the VA, VB and AS-type neurons. In these neurons, the onset of cholinergic marker expression is observed already in late L1 stage animals, contrasting the late L4 onset in the VC neurons (*Figure 5C*).

Other than the CEM neurons, there are no sex-specific neurons located in the head of the worm. We were therefore surprised to note a pair of neurons, located next to the cholinergic AIY interneurons in the ventral head ganglion that expressed cholinergic markers only in males, but not hermaphrodites (*Figure 5D*). This neuron pair is the AIM neuron pair, previously implicated in olfactory memory formation (*Lakhina et al., 2015*) and mate searching behavior (*Barrios et al., 2012*). In hermaphrodites, the AIM neurons are glutamatergic, expressing the vesicular glutamate transporter *eat-4/VGLUT* (*Figure 5D*). In males, the AIM neurons also initially express *eat-4/VGLUT*, but only

**Table 5.** Male-specific cholinergic neurons.

| Neuron type | Neuron class | unc-17 fosmid expression | cho-1 fosmid expression | Co-transmitter | Previous ID |
|---|---|---|---|---|---|
| Sensory neuron (7 classes) | CEM D/V L/R | ++ | ++ | | no |
| | R1A, R2A, R3A, R4A, R6A | ++ | ++ | | yes [1] |
| | SPC L/R | ++ | ++ | | yes [2] |
| | SPV L/R | ++ | ++ | | yes [3] |
| | HOB | +++ | ++ | | no |
| | PCB L/R | ++ | ++ | | yes [2] |
| | PCC L/R | ++ | ++ | | no |
| Interneuron (6 classes) | DVE | ++ | | | no |
| | DVF | ++ | | | no |
| | PDC | ++ | ++ | PDC or PGA are also serotonergic [4] | no |
| | PGA | ++ | | PDC or PGA are also serotonergic [4] | no |
| | PVY | +++ | ++ | | yes[3] |
| | PVX | +++ | ++ | | yes[3] |
| Motor neuron (3 classes) | PVZ | +++ | ++ | | no |
| | PVV | +++ | ++ | Glutamatergic[6] | no |
| | CA1-9* | ++ | ++ | | no[7] |

*CA7-9 do not express cho-1 and have lower levels of unc-17 than CA1-6.

[1] **Koo et al. (2011)**.

[2] **Garcia et al. (2001)**.

[3] **LeBoeuf et al. (2014)**.

[4] **Loer and Kenyon (1993)**.

[5] **Sherlekar et al. (2013)**.

[6] Our unpublished data.

[7] Rand and Nonet cite unpublished observations of cholinergic identity of four CA neurons (**Rand and Nonet, 1997**). We observe expression of unc-17 in all nine CA neurons (albeit lower in CA7-9).

until the L3 stage. During the L4 stage eat-4/VGLUT expression becomes downregulated and unc-17/VAChT and cho-1/ChT expression becomes induced (**Figure 5E**).

We assessed whether the neurotransmitter switch of the AIM neurons is programmed in a cell autonomous manner. To this end, we generated sexually mosaic animals in which we masculinized AIM in otherwise hermaphroditic animals and we feminized AIM in otherwise male animals, using previously described strategies (**Lee and Portman, 2007**; **Mowrey et al., 2014**; **White and Jorgensen, 2012**; **White et al., 2007**). Specifically, masculinization was achieved by degrading the global regulator of hermaphroditic cellular identity, TRA-1, by ectopic expression of FEM-3 in specific hermaphroditic cells; FEM-3 is normally functioning in males to globally degrade TRA-1. Feminization is achieved by preventing FEM-3 downregulation of TRA-1 in male cells through ectopic expression of the intracellular domain of TRA-2 (TRA-2ic), which normally acts in hermaphrodites to inhibit FEM-3. FEM-3 or TRA-2ic were expressed under a fragment of the eat-4 locus, which is exclusively expressed in the AIM neurons in the head ganglia of the worm (E.S. and O.H., unpubl.). We found that masculinization of the AIM neurons ('eat-4prom11::fem-3') in otherwise hermaphroditic animals results in downregulation of eat-4/VGLUT and upregulation of cho-1 expression (**Figure 5F**). Conversely, feminization of AIM in male animals results in sustained eat-4 expression and no induction of cho-1 expression (**Figure 5F**). These results demonstrate that the neurotransmitter switch is programmed cell autonomously.

## Transcriptional control of the cholinergic neurotransmitter phenotype

Neurotransmitter maps can serve many different purposes. One of their applications relates to nervous system development. Since the neurotransmitter identity of a neuron defines a critical identity

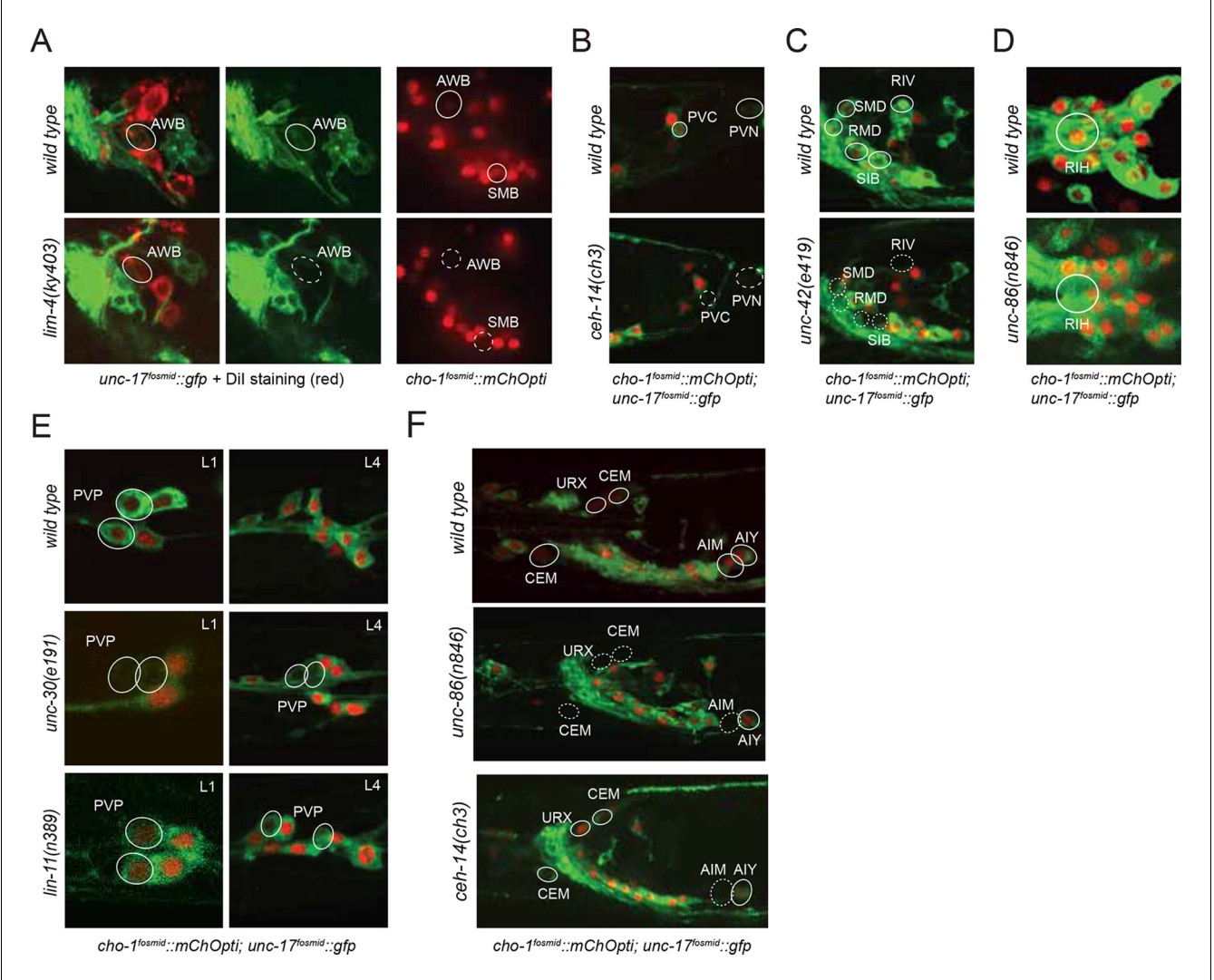

**Figure 6.** Regulatory factors affecting cholinergic identity. We examined 20 animals for each genotype and for every mutant strain the described phenotype was observed in >80% of animals. (**A**) The LIM homeobox transcription factor *lim-4* is required for *unc-17* fosmid reporter expression (left panel) and *cho-1* fosmid reporter expression (right panel) in AWB and SMB neurons . AWB neurons were visualized by DiI staining in the *unc-17* fosmid reporter expressing strain. AWB and SMB show no fosmid reporter expression in the *lim-4* mutant. (**B**) The Otx-type homeobox transcription factor *ceh-14* is required for *unc-17* and *cho-1* fosmid reporter expression in PVC and *unc-17* fosmid reporter expression in PVN. PVC neurons show a decrease in *unc-17* and *cho-1* fosmid reporter expression in the *ceh-14* mutant compared to *wild type*. PVN neurons show no *unc-17* fosmid reporter expression in the *ceh-14* mutant. Note that PVN does not express *cho-1* fosmid reporter in wild type animals. (**C**) The homeobox transcription factors *unc-30* and *lin-11* are required for normal expression of the *unc-17* and *cho-1* fosmid reporters. Cholinergic identity genes are downregulated in PVP neurons starting at L1 (top panels) and continuing until the L4/adult stage (bottom panels) in *unc-30* and *lin-11* mutant strains compared to *wild type*. (**D**) The homeobox transcription factor *unc-42* is required for *unc-17* and *cho-1* fosmid reporter expression in RIV, SMD, RMD and SIB. (**E**) The POU homeobox transcription factor *unc-86* is required for *unc-17* and *cho-1* fosmid reporter expression in RIH. (**F**) A *wild type* male is shown in the top panel for reference. *unc-86* (middle panel) is also required for *unc-17* and *cho-1* fosmid reporter expression in URX and in the CEM male-specific neurons. In the absence of *unc-86* the AIM neurons did not show expression of *unc-17* and *cho-1* fosmid reporters in the L4/adult male. The LIM homeobox transcription factor *ceh-14* is required for the AIM neurons to express *unc-17* and *cho-1* fosmid reporters in the L4/adult male (bottom panel). Transgenes: *otIs576* = *unc-17* fosmid reporter; *otIs544* = *cho-1* fosmid reporter.

The following figure supplement is available for figure 6:

**Figure supplement 1.** Continuous expression of transcription factors fosmid reporters in cholinergic neurons.

feature of any specific neuron type, a neurotransmitter map provides an entry point to study the molecular mechanisms by which neuronal identity is acquired. Previous work from our lab has defined transcription factors that control cholinergic identity in a small number of sensory, inter- and motor neurons. Specifically, we have reported that the POU homeobox gene *unc-86* controls the cholinergic identity of three cholinergic sensory neurons (IL2, URA, URB) (*Zhang et al., 2014*), that the LIM homeobox gene *ttx-3* controls the cholinergic identity of two cholinergic interneurons (AIY, AIA) (*Altun-Gultekin et al., 2001*) and that the COE (Collier/Olf/EBF)-type Zn-finger factor *unc-3* controls cholinergic identity of most motor neuron classes in the VNC as well as the SAB head motor neurons (*Kratsios et al., 2011*, *2015*). We sought to extend this analysis to other neuron classes, with the specific question in mind whether broad themes of neurotransmitter identity control may be revealed through the establishment of a comprehensive 'regulatory map'. To identify transcriptional regulators, we examined candidate factors known to be expressed in specific neurons and also conducted genetic screens using *gfp*-based identity markers of cholinergic neurons (see Materials and methods). Our analysis resulted in the identification of a total of 7 regulators that control the identity of 20 of the 52 cholinergic neuron types (*Table 6*; *Figure 6*).

In line with a similar observation that we made upon analysis of glutamatergic neuron identity control (*Serrano-Saiz et al., 2013*), we observed a striking preponderance of homeodomain containing proteins in the transcription factors that we newly identified as cholinergic identity regulators. Specifically, we found that the three LIM homeobox genes *lim-4, lin-11* and *ceh-14* control cholinergic identity of six distinct cholinergic neuron types, including sensory neurons (*lim-4* in AWB), interneurons (*lin-11* in PVP; *ceh-14* in AIM, PVC) and motor neurons (*ceh-14* in PVN, *lim-4* in SMB; *Table 6*; *Figure 6*). However, we find that *lim-7*, the *C. elegans* homolog of vertebrate Islet, which specifies cholinergic identity in the spinal cord and forebrain in mice (*Cho et al., 2014*), is not required to specify cholinergic identity in *C. elegans* (as assessed by normal *cho-1* expression throughout the nervous system in *lim-7* null mutants; data not shown). Therefore, while the usage of LIM-type homeobox genes in controlling cholinergic neurotransmitter identity appears to be conserved from *C. elegans* to vertebrates, different family members appear to execute this function in different species and cell types.

Moreover, we found that the Pitx-type homeobox gene *unc-30* controls cholinergic identity of the PVP interneurons (in conjunction with *lin-11*) and that the POU homeobox gene *unc-86* controls cholinergic identity of the URX, RIH and male-specific CEM neurons. *unc-86*, in conjunction with *ceh-14*, is also required for the AIM neurons to adopt their cholinergic identity in males; both factors also control glutamatergic identity of the AIM neurons in hermaphrodites (and males till the third larval stage; *Figure 6*). All of the above-mentioned transcription factors are continuously expressed throughout the life of these neurons (*Figure 6—figure supplement 1*), suggesting that these factors not only initiate but also maintain cholinergic identity.

From a EMS-induced genetic mutant screen that we conducted for regulators of RMD neuron identity (see Materials and methods), we uncovered *unc-42*, a Prox-type homeobox gene as a regulator of cholinergic gene expression in RMD motor neurons. We also found that *unc-42* affects cholinergic identity of four additional, distinct types of cholinergic head neurons, most of them motor neurons (*Figure 6*). *unc-42* is continuously expressed in all these postmitotic neuron types (*Figure 6—figure supplement 1*).

The only exception to the homeobox theme is what appears to be the most remarkable regulator of cholinergic identity, the phylogenetically conserved COE-type *unc-3* transcription factor. In addition to the previously reported impact of *unc-3* on cholinergic ventral cord motor neuron identity (SAB-, A-, B-, AS-type MNs), we found that *unc-3* regulates expression of the cholinergic identity genes *cho-1* and *unc-17* in all command interneurons (*Figure 7*, *Table 6*). Moreover, the PDA, PDB, PVN and DVA tail neurons also require *unc-3* for their normal expression of cholinergic identity genes (*Figure 7*, *Table 6*). DVA is particularly notable here because like the command interneurons, the DVA neuron also takes a central role in the overall *C. elegans* connectivity network (*Varshney et al., 2011*) (*Figure 3B*) and this central location is paralleled by the dependence of these neurons on *unc-3* activity. The expression pattern of *unc-3* had not previously been reported in most of these neurons. Using a fosmid reporter and a *gfp* reporter inserted into the *unc-3* locus through CRISPR-Cas9, we confirmed expression of *unc-3* in all these cholinergic neuron types, including the command interneurons (*Figure 7A*).

**Table 6.** Newly identified transcriptional regulators of cholinergic identity.

| Gene* | DNA binding domain | Neuron class | Effect on identity features | | | Other neurotransmitter identities affected (neuron class) |
|---|---|---|---|---|---|---|
| | | | Cholinergic identity** | | Other identity features** | |
| | | | unc-17 cha-1 | cho-1 | | |
| unc-3 (EBF) | Zn finger | PDA | yes | yes | yes | |
| | | PDB | yes | yes | n.d. | |
| | | DVA | yes | yes | no | |
| | | PVC | yes | yes | no | |
| | | AVA | yes | yes | no | |
| | | AVB | yes | yes | no | |
| | | AVD | yes | yes | no | |
| | | AVE | yes | yes | no | |
| | | PVN | yes | n.a. | n.d. | |
| unc-42 (Prd-type) | Homeodomain | RIV | yes | yes | n.d. | Glu(ASH) [7] |
| | | RMD | yes | yes | yes | |
| | | SMD | yes | yes | n.d. | |
| | | SIB | yes | yes | n.d. | |
| | | AVA | no | no | yes[1] | |
| | | AVD | no | no | yes[1] | |
| | | AVE | no | no | yes[1] | |
| lim-4 (Lhx6/8) | Homeodomain | AWB | yes | yes | yes[2] | |
| | | SMB | yes | yes | yes[3] | |
| | | RIV | no | no | n.d. | |
| lin-11 (Lhx1) | Homeodomain | ADF | no | no | no | Glu (ASG, ADL) [7] |
| | | PVP | yes | yes | yes[4] | |
| unc-30 (Pitx) | Homeodomain | PVP | yes | yes | yes[4] | GABA (DD, VD) [9] |
| | | RIH | no | no | n.d. | |
| unc-86 (Brn3) | Homeodomain | CEM (male) | yes | yes | yes[5] | Glu (ALM, PLM, AIM, AIZ, AQR, PQR, PVR) [7] |
| | | URX | yes | yes | yes[6] | |
| | | AIM (male) | yes | yes | yes[7] | |
| | | RIH | yes | yes | yes[8] | |
| ceh-14 (Lhx3/4) | Homeodomain | AIM (male) | yes | yes | yes | Glu (AFD, DVC, PHA, PHB, PHC) [7] |
| | | PVN | yes | yes | n.d. | |
| | | PVC | yes | yes | yes | |

*Vertebrate orthologs in parenthesis. All neuron classes listed express the respective transcription factor tested.

**'yes' = expression is downregulated or completely absent; 'no' = no readily observable effect; 'n.d.' = not determined; 'n.a.' = not applicable because gene is not expressed in this cell. For primary data see **Figure 6**, and **Figure 8**. For data on 'other markers' (≥2 markers tested), see individual footnotes (this data is partly our own data, partly previously reported data). .Previously identified regulators of cholinergic identity are: unc-3 in A-, B-type, AS and SAB motor neurons, unc-86 in IL2, URA, URB, cfi-1 in IL2, URA, ttx-3 in AIY and AIA and ceh-10 in AIY (**Altun-Gultekin et al., 2001**; **Kratsios et al., 2011**, **2015**; **Wenick and Hobert, 2004**; **Zhang et al., 2014**).

[1] **Baran et al. (1999)**; **Brockie et al. (2001)**.

[2] **Alqadah et al. (2015)**; **Sagasti et al. (1999)**.

[3] **Kim et al. (2015)**.

[4] **Hutter (2003)**.

[5] **Shaham and Bargmann (2002)**.

[6] **Qin and Powell-Coffman (2004)**.

[7] **Serrano-Saiz et al. (2013)**.

[8] **Sze et al. (2002)**.

[9] **McIntire et al. (1993**)

Apart from the preponderance of homeobox genes, another striking theme we found is the employment of the same transcription factor in completely different cellular contexts, apparently a reflection of the operation of transcription factors in distinct combinations. For example, *unc-86* controls cholinergic identity in the IL2 sensory neurons and the unrelated AIM interneurons (in the male). In these different cellular contexts *unc-86* cooperates with distinct cofactors, *cfi-1* in IL2 (*Zhang et al., 2014*) and *ceh-14* in AIM (*Figure 6G*). The need for specific combinations of transcription factors to drive a specific identity program explains why we find that a factor that is expressed in multiple cholinergic neuron types does not necessarily regulate cholinergic identity in all neuron types in which it is expressed (*Table 6*). For example, *lim-4* which is expressed in the cholinergic AWB, SMB and RIV neurons controls cholinergic identity in AWB and SMB (*Figure 6*), but not in RIV. This is likely because the cofactors that work together with *lim-4* in AWB and/or SMB may not be expressed in RIV. Likewise, *ceh-10* forms a heterodimer with *ttx-3* in AIY to control its cholinergic identity (*Altun-Gultekin et al., 2001*; *Wenick and Hobert, 2004*), but it is not required for cholinergic identity of the AIN neurons which express *ceh-10*, but not *ttx-3* (data not shown).

Transcription factors that we find to control cholinergic neurotransmitter identity are also employed in the control of other neurotransmitter identities (see *Table 6*), likely in the context of distinct transcription factors combinations. For example, the Pitx-type homeobox gene *unc-30* controls cholinergic identity of the PVP neurons, apparently in conjunction with *lin-11* (this study), but also controls the GABAergic identity of D-type VNC MNs, likely in conjunction with an as yet unidentified factor (*Jin et al., 1994*). Likewise, the LIM homeobox gene *ceh-14*, which controls cholinergic identity of the PVN and PVC neurons (*Table 6*), likely together with *unc-3* (this study), controls glutamatergic identity of various amphid and phasmid sensory neurons in which *ceh-14* operates independently of *unc-3* (*Serrano-Saiz et al., 2013*).

Lastly, we note that loss of two of the transcription factors that we examined, *unc-3* and *ceh-14*, results in derepression of cholinergic identity features in normally non-cholinergic neurons (data not shown). In *unc-3* mutants, two cells in the dorsal ganglion ectopically express cholinergic features; these are probably the RID neuron and its sister cell. In *ceh-14* mutants at least one pair of tail neurons ectopically expresses cholinergic markers.

## Coupling of cholinergic identity with other terminal identity features

Most of the transcriptional regulators that we defined here control not only cholinergic identity in the respective neuron classes, but also control other identity features. For example, we find that loss of *unc-3* affects multiple aspects of PDA motor neuron identity (expression of the *exp-1* ligand gated ion channel, *ace-3/4* cholinesterase, *cog-1* homeobox gene) and loss of *unc-42* affects metabotropic glutamate receptor (*mgl-1*) expression in the RMD neurons. Apart from affecting cholinergic identity, loss of *ceh-14* affects neuropeptide (*flp-10*) expression, as well as the serotonergic co-transmitter identity of the AIM neurons and it affects expression of the ionotropic glutamate receptors *nmr-1* and *glr-1* in the PVC command interneuron (*Figure 8*). In addition, *lin-11* and *unc-30* were previously found to control many terminal identity features of the PVP (*Hutter, 2003*) and these two factors also control cholinergic identity of PVP (*Figure 6E*). Similarly, *lim-4* controls cholinergic identity of the AWB neurons but also several other identity features (*Alqadah et al., 2015*; *Sagasti et al., 1999*). The coupling of adopting cholinergic identity control with the adoption of other identity features has been observed in previously described regulators of cholinergic identity: *unc-3* for VNC MNs (*Kratsios et al., 2015 2011*), *unc-86* for IL2 (*Zhang et al., 2014*), *ttx-3* for AIY and AIA (*Altun-Gultekin et al., 2001*; *Zhang et al., 2014*), and also in the context of neurons with distinct neurotransmitter identities (e.g. [*Flames and Hobert, 2009*; *Serrano-Saiz et al., 2013*]).

However, we also noted a number of striking exceptions to the coupling of neurotransmitter identity with other terminal identity features. The serotonergic identity of the hermaphrodite specific motor neurons HSN is controlled by the *unc-86* POU homeobox gene (*Sze et al., 2002*), but *unc-86* does not affect *unc-17/VAChT* expression in HSN (data not shown). The most striking example for a separation of neurotransmitter identity from other identity features is observed in relation to the function of the *unc-3* gene. We had previously shown that in all motor neurons in which *unc-3* is expressed (SAB head motor neurons, A- B- and AS-type VNC MNs), *unc-3* not only controls neurotransmitter identity, but also a multitude of other terminal molecular markers (*Kratsios et al., 2015, 2011*). In contrast, the activity of *unc-3* in the AVA, AVB, AVD, AVE and PVC command interneurons and DVA interneuron appears to be restricted to select subfeatures of these neurons. We arrived at

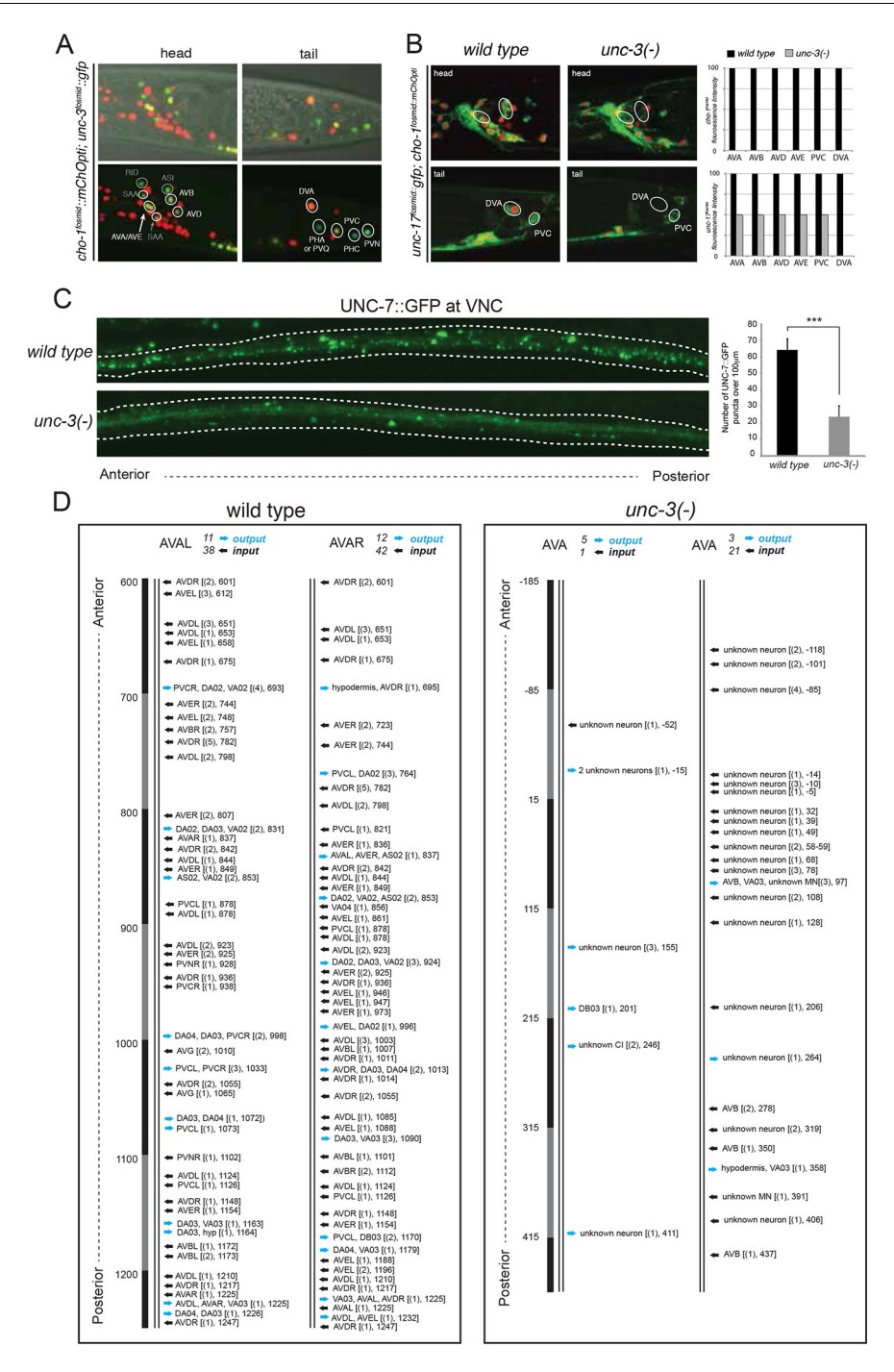

**Figure 7.** *unc-3* is a circuit-associated transcription factor. (**A**) Expression pattern of an *unc-3* fosmid-based reporter (*otls591*). Overlap with a *cho-1* fosmid-based reporter (*otls544*) is shown in all panels. The upper panels are the same as the lower, but a Nomarski image has been added for orientation purposes. *unc-3* expression was also detected in PDA, PDB and PVP in the pre-anal ganglion (data not shown). (**B**) The expression of the *unc-17* and *cho-1* fosmid reporters is downregulated in command interneurons (AVA, AVB, AVD, AVE, PVC) and the tail neuron DVA in *unc-3* mutant animals (identical results were obtained using two *unc-3* alleles, *e151* generates a premature STOP and *n3435* is a deletion allele). Quantification is shown on the right. Twenty animals were analyzed at the fourth larval stage (L4) per genotype. Note that the effect of *unc-3* on *unc-17* expression in the command interneurons (this figure) is not as fully penetrant as it is in VNC motor neurons (*Kratsios et al., 2011*). (**C**) Gap junctions that command interneurons make are visualized with *gfp* tagging the innexin protein UNC-7, as previously described (*Starich et al., 2009*) (transgene: *iwls47*). Dotted white lines delineate the location of the VNC. A significant decrease in the number of the UNC-7::GFP puncta was observed in the VNC of *unc-3(n3435)* mutant animals (quantification shown on the right with average values and standard deviation). A student's t test was performed. ***p value <0.0001.
*Figure 7 continued on next page*

*Figure 7 continued*

(D) Reconstruction of the chemical synapse connectivity of the AVA command interneurons in a wild type and an *unc-3(e151/MnH205)* mutant animal. Less synaptic input onto AVA neurons and output from the AVA neurons was observed in the *unc-3* mutant animal. This is not merely an effect of axonal process misplacement since in *unc-3* mutants, AVA processes still run adjacent to the processes of the neurons it normally makes synaptic contacts to. More than 600 electron micrographs were reconstructed per genotype. In square brackets, the location (number of electron micrograph) for each chemical synapse is shown, and the number of consecutive micrographs in which a synapse was detected is also shown in parenthesis.

The following figure supplement is available for figure 7:

**Figure supplement 1.** UNC-3 has no effect on glutamate receptor expression in command interneurons.

this conclusion by analyzing the expression of more than ten additional identity markers of these *unc-3*-expressing neurons (including glutamate receptors, neuropeptides and ion channels). Not a single one besides the cholinergic reporter genes is affected in the worms lacking *unc-3* (**Figure 7— figure supplement 1**; **Table 7**). Within a subset of these neurons, namely the command interneurons AVA, AVD and AVE, three transcription factors, *unc-42* (homeobox), *fax-1* (nuclear hormone receptor) and *cfi-1* (ARID-type), have been shown to control subsets of these *unc-3*-independent terminal identity markers (**Table 8**). The observation of a piece-meal regulation of distinct terminal identity features by a number of distinct transcription factors, each acting in a highly cell-type and target gene-specific manner (**Table 8**), represents a remarkable departure from the commonly observed theme of co-regulation of multiple identity features by the same set of transcription factors

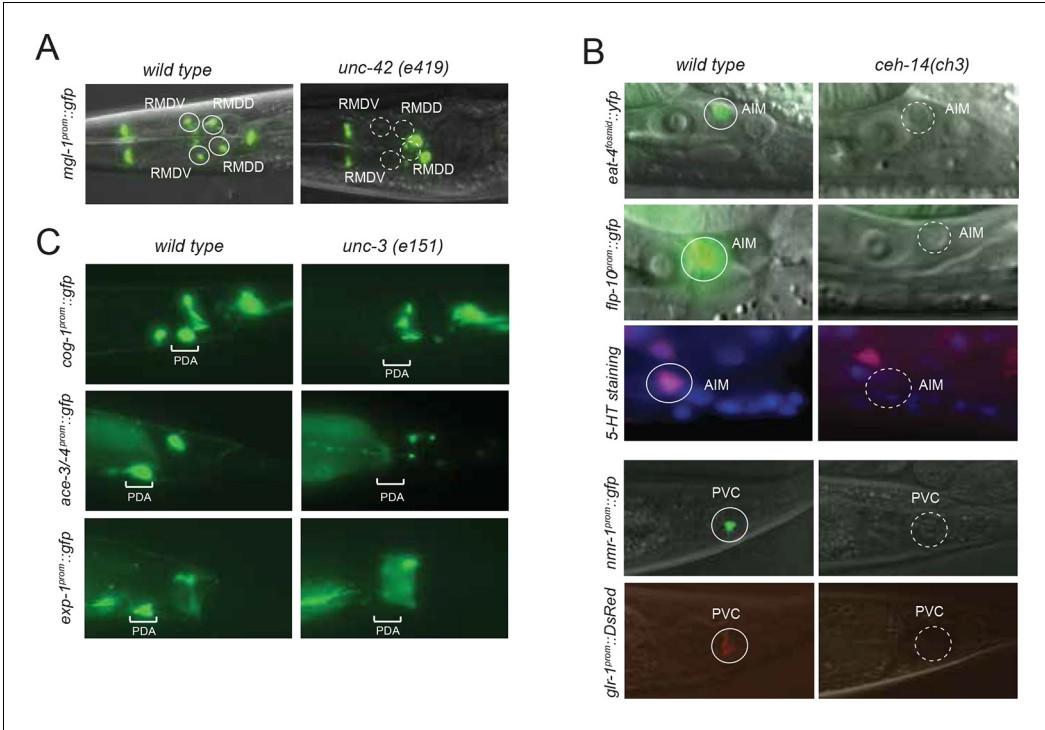

**Figure 8.** Coupling of cholinergic identity with other identity features. (**A**) An *mgl-1* reporter transgene does not show expression in RMD neurons in the absence of *unc-42*. (**B**) In the absence of *ceh-14* the AIM neurons do not show *eat-4* fosmid reporter and *flp-10* reporter expression. 5-HT staining is not detectable in AIM neurons in the *ceh-14* mutant. In the absence of *ceh-14* the PVC neurons do not show *nmr-1* or *glr-1* reporter expression. Number of animals examined = 20 animals per reporter gene per genotype. (**C**) The expression of PDA identity markers *exp-1*, *ace-3/4*, and *cog-1* is lost in *unc-3* mutant animals. For *cog-1prom::gfp*, 25 of 25 wild-type and 1 of 25 *unc-3(e151)* animals showed *cog-1prom::gfp* expression in PDA. For *ace-3/-4prom::gfp*, 20 of 20 wild-type and 0 of 20 *unc-3(e151)* animals showed *ace-3/-4prom::gfp* expression in PDA. For *exp-1prom::gfp*, 20 of 20 wild-type and 11 of 20 *unc-3(e151)* animals showed *exp-1prom::gfp* expression in PDA.

**Table 7.** *unc-3* affects the differentiation of a broad set of cholinergic neuron types. *nmr* and *glr* genes encode glutamate receptors and expression of neither is affected by *unc-3*; many of them are instead regulated by either the *unc-42, fax-1 or cfi-1*, as summarized in **Table 8**.

| | | *unc-3* (+) neurons | Cholinergic identity in *unc-3(-)* animals [1] | Other identity features that are NOT affected in *unc-3(-)* animals [1] | Other identity feature that are affected in *unc-3(-)* animals |
|---|---|---|---|---|---|
| INTER | Command inter-neurons | AVA | *unc-17, cho-1* affected | *nmr-1, nmr-2, glr-1, glr-2, glr-4, glr-5, acr-15, rig-3, flp-18* | |
| | | AVB | *unc-17, cho-1* affected | *acr-15* | |
| | | AVD | *unc-17, cho-1* affected | *nmr-1, nmr-2, glr-1, glr-2, glr-5* | |
| | | AVE | *unc-17, cho-1* affected | *nmr-1, nmr-2, glr-1, glr-2, glr-5, opt-3* | |
| | | PVC | *unc-17, cho-1* affected | *nmr-1, nmr-2, glr-1, glr-2, glr-5* | |
| | Other inter-neurons | DVA | *unc-17, cho-1* affected | *glr-4, glr-5, twk-16, nlp-12, zig-5, ser-2* | |
| | | SAA | *unc-17* NOT affected[2] | | |
| | | PVP | *unc-17, cho-1* NOT affected | | |
| MOTOR | Head MNs | SAB | *unc-17, cho-1* affected[3] | | 8/8 markers tested[3] |
| | VNC MNs | A,B,AS | *unc-17, cho-1* affected[3] | | 29/34 markers tested[3] |
| | Tail MNs | PDA | *unc-17, cho-1* affected | | *exp-1, ace-3/4, cog-1* [1] |
| | | PDB | *unc-17, cho-1* affected | | |
| | | PVN | *unc-17* affected[2] | | |

[1]See **Figure 7**, **Figure 7—figure supplement 1**
[2]Cho-1 not expressed in these neurons.
[3]As previously reported by **Kratsios et al. (2011)**, **(2015)**.

(**Alqadah et al., 2015**; **Cinar et al., 2005**; **Duggan and Chalfie, 1995**; **Etchberger et al., 2009**; **Guillermin et al., 2011**; **Hobert, 2011**; **Kratsios et al., 2011**; **Serrano-Saiz et al., 2013**; **Wenick and Hobert, 2004**; **Zhang et al., 2014**).

## *unc-3* is a circuit-associated transcription factor

Moving beyond a cell- and gene-centric consideration of regulatory factors, we asked whether there are any overarching, circuit-based themes of cholinergic identity control. Specifically, since every transcription factor that we identified here to control cholinergic neurotransmitter identity exerts its effect on more than one neuron type, we asked whether neurons whose neurotransmitter identity is controlled by the same regulatory factor are part of synaptically connected circuits. Such an observation would suggest that the respective transcription factor may define and coordinate the activity of entire circuits and perhaps may also define aspects of circuit assembly. We indeed found several examples of transcription factors that control the identity of synaptically connected neurons.

The most striking example is the ventral nerve cord motor circuit which is composed of a multitude of interconnected motor neurons (six different classes) and a highly interconnected 'rich club' of interneurons (also six different classes) (**White et al., 1986**; **Towlson et al., 2013**). As noted above, the entire ventral nerve cord motor circuit uses ACh (except DD/VD; shown again in **Figure 9A**). Strikingly, *unc-3* is expressed and required for the adoption of cholinergic identity in all neurons in this circuit (schematized in **Figure 9A**; data in **Figure 7B**). *unc-3* is therefore a circuit-associated transcription factor that is selectively associated with this circuit (*unc-3* is expressed only in a few neurons outside this circuit) and that defines a critical feature of the circuit, namely the ability of neurons in the circuit to communicate among each other. We furthermore note that the regulated mutual 3-neuron network motif mentioned above (**Figure 4E**) frequently occurs in the *unc-3*-dependent motor circuit, with the mutually connected neurons being *unc-3*-dependent command interneurons that receive inputs either from glutamatergic neurons outside the circuit or from cholinergic, and also *unc-3*-dependent neurons within the circuit (**Figure 9A**; **Table 4**).

To investigate whether neurotransmitter identity is the only parameter of the circuit that is disrupted in *unc-3* mutants, we examined connectivity between neurons in the VNC MN circuit. In our

**Table 8.** Transcription factors affecting command interneuron differentiation.

| TF | AVA | AVB | AVD | AVE | PVC |
|---|---|---|---|---|---|
| *unc-3* | | | | | |
| *unc-42* | | | | | |
| *fax-1* | | | | | |
| *cfi-1* | | | | | |
| *ceh-14* | | | | | |
| **ACh** | | | | | |
| *unc-17 & cho-1* | *unc-3* effect | *unc-3* effect | *unc-3* effect | *unc-3* effect | *unc-3* effect |
| | *unc-42* NO effect | | *unc-42* NO effect | *unc-42* NO effect | |
| | | | | | *ceh-14* effect |
| **Other** | | | | | |
| *nmr-1* (GluR) | *unc-3* NO effect | | *unc-3* NO effect | *unc-3* NO effect | *unc-3* NO effect |
| | *unc-42* NO effect | | *unc-42* NO effect | *unc-42* NO effect | *ceh-14* effect |
| | *fax-1* effect | | *fax-1* NO effect | *fax-1* effect | |
| | | | *cfi-1* effect | | *cfi-1* effect |
| *nmr-2* (GluR) | *unc-3* NO effect | | *unc-3* NO effect | *unc-3* NO effect | *unc-3* NO effect |
| | *unc-42* NO effect | | *unc-42* NO effect | *unc-42* NO effect | |
| | *fax-1* effect | | *fax-1* NO effect | *fax-1* effect | |
| *glr-1* (GluR) | *unc-3* NO effect | *unc-3* NO effect | *unc-3* NO effect | *unc-3* NO effect | *unc-3* NO effect |
| | *unc-42* effect | *unc-42* NO effect | *unc-42* effect | *unc-42* effect | *ceh-14* effect |
| | *fax-1* no effect | *fax-1* NO effect | *fax-1* no effect | *fax-1* no effect | |
| | | | *cfi-1* effect | | *cfi-1* effect |
| *glr-2* (GluR) | *unc-3* NO effect | | *unc-3* NO effect | *unc-3* NO effect | *unc-3* NO effect |
| | *unc-42* NO effect | | *unc-42* NOeffect | *unc-42* NOeffect | |
| | *fax-1* NO effect | | *fax-1* NO effect | *fax-1* NO effect | |
| *glr-4* (GluR) | *unc-3* NO effect | | | | |
| | *unc-42* effect | | | | |
| | *fax-1* no effect | | | | |
| *glr-5* (GluR) | *unc-3* NO effect | *unc-3* NO effect | *unc-3* NO effect | *unc-3* NO effect | *unc-3* NO effect |
| | *unc-42* effect | *unc-42* NOeffect | *unc-42* effect | *unc-42* effect | |
| | *fax-1* no effect | *fax-1* NO effect | *fax-1* no effect | *fax-1* no effect | |
| *opt-3* | | | | *unc-3* NO effect | |
| | | | | *unc-42* effect | |
| | | | | *fax-1* effect | |
| *rig-3* (IgSF) | *unc-3* no effect | | | | |
| *flp-18* (FMRF) | *unc-3* no effect | | | | |

Gray shading: gene normally expressed in this cell. 'Effect' (red) indicate that respective reporter gene fails to be expressed in the respective mutant background in the indicated cells, 'no effect' (green) indicates the opposite.

*unc- 42*, *cfi-1* and *fax-1* data on non-ACh marker from **Wightman et al. (2005)** **Shaham and Bargmann (2002)** and **Brockie et al. (2001)**

previous analysis of *unc-3* function, we had identified neuromuscular junction defects, i.e. disorganized or absent synapses from VNC MNs onto body wall muscle (**Kratsios et al., 2015**), but upstream layers of the motor circuit (i.e. connections of command interneurons to MNs and connections among command interneurons) had not been examined. *unc-3*-expressing command interneurons make prominent electrical synapses to other command interneurons and to motor neurons and

these synapses can be visualized through *gfp*-tagging of a gap junction component that connects command interneurons and motor neurons, the innexin *unc-7* (*Figure 7C*), which is expressed in command interneurons (*Starich et al., 2009*). UNC-7::GFP puncta, visualized with a translational reporter are severely reduced in *unc-3* mutants (*Figure 7C*). Expression of a transcriptional, fosmid-based *unc-7* reporter is unaffected in *unc-3* mutants (data not shown), leading us to conclude that *unc-3* affects electrical synapse formation at a step independent of regulation of innexin expression.

To examine chemical synapses within neurons of the motor circuit, we reconstructed the chemical synapse connectivity of the AVA command interneuron in *unc-3* null mutants using serial analysis of electron micrographs. We reconstructed a defined part of the anterior ventral nerve cord between two different motor neurons (AS1 and AS3). In this region, AVA makes prominent chemical synapses onto MNs and other command interneurons and it receives several synaptic inputs (*Figure 7D*). In *unc-3* null mutants, we found connectivity defects on all levels: AVA receives less chemical synaptic input from within the motor circuit (i.e. from other command interneurons) and it makes less chemical synapses onto other motor neurons and onto other command interneurons (*Figure 7D*). There is also an overall disorganization of the placement of axonal processes in the VNC of *unc-3* mutants (data not shown); however, AVA still neighbors the command interneurons that it normally connects to, indicating that the connectivity defects are not a secondary consequence of placement defects.

Non-cholinergic synaptic inputs from sensory neurons into the motor circuit appear not to be affected by *unc-3*. We arrived at this conclusion by examining the synaptic connections of the glutamatergic PHB neuron to the AVA interneuron, normally made in the pre-anal ganglion (*White et al., 1986*). This synaptic connection can be visualized using a GFP reconstitution system ('GRASP'; (*Park et al., 2011*). We find these synaptic GFP signals to be unaffected in *unc-3* mutants (data not shown).

Remarkably, the pan-circuit control of cholinergic neurotransmitter identity by *unc-3* is mediated via a single UNC-3 binding site ('COE motif') controlling neurotransmitter pathway genes. Its deletion in the context of the *cho-1*/ChT *fosmid*-based reporter eliminates expression not only in the ventral nerve cord motor neurons, but also in all other *unc-3* dependent cholinergic neurons, not just within the motor neuron circuit, but also outside the circuit (*Figure 10A–C*). On the other hand, a 280 bp region from the *cho-1* and a 250 bp region from the *unc-17* locus that contain the COE motif are not sufficient to drive expression in all *unc-3*-dependent inter- and motor neurons of the motor circuit, but only drives expression in motor neurons (*Figure 10D, E*). This finding suggests that *unc-3* may cooperate with distinct cofactors in distinct neuron types.

Taken together, *unc-3* activity is required not only for the expression of proper neurotransmitter identity, but also for synaptic connectivity throughout the VNC motor neuron circuit, not just in motor neurons but also in command interneurons. However, as mentioned above, *unc-3* is not required to control the expression of other identity features of command interneurons, such as the many types of distinct glutamate receptors expressed by the command interneurons (*Figure 7—figure supplement 1*).

## *unc-42* appears to be another circuit-associated transcription factor

*unc-3* may not be the only transcription factor whose activity is required for the function and assembly of an entire circuit. On a micro-circuit level, we note that the homeobox gene *unc-42* is, like *unc-3,* also frequently employed in the context of the 'regulated mutual' 3-neuron network motif described above. This motif is predominantly found either (a) in the context of the innervation of cross-connected command interneurons or (b) the context of cross-connected head motor neurons (SMDs and RMDs; *Figure 9B*; *Table 4*). *unc-42* has functions in both of these motifs. In the case of the cross-connected head motor neurons (RMDs, SMDs) and the RIV interneuron that innervates these cross-connected neurons, *unc-42* specifies the cholinergic identity of all of these neurons (and other signaling input to these neurons, exemplified by the above-mentioned regulation of the metabotropic Glu receptor *mgl-1* by *unc-42*). In the cross-connected command interneurons, *unc-42* does not affect their cholinergic identity, but it does affect the expression of multiple ionotropic Glu receptors (GluRs) expressed in the command interneurons (*Brockie et al., 2001*). Intriguingly, in a number of motif occurrences, the cross-connected command interneurons are innervated by the glutamatergic ASH sensory neurons, which sense a number of repulsive cues (*Table 4*) (*Kaplan and Horvitz, 1993*). We previously found that the glutamatergic identity of the ASH neurons is controlled by *unc-42* (*Serrano-Saiz et al., 2013*). *unc-42* therefore controls and apparently coordinates

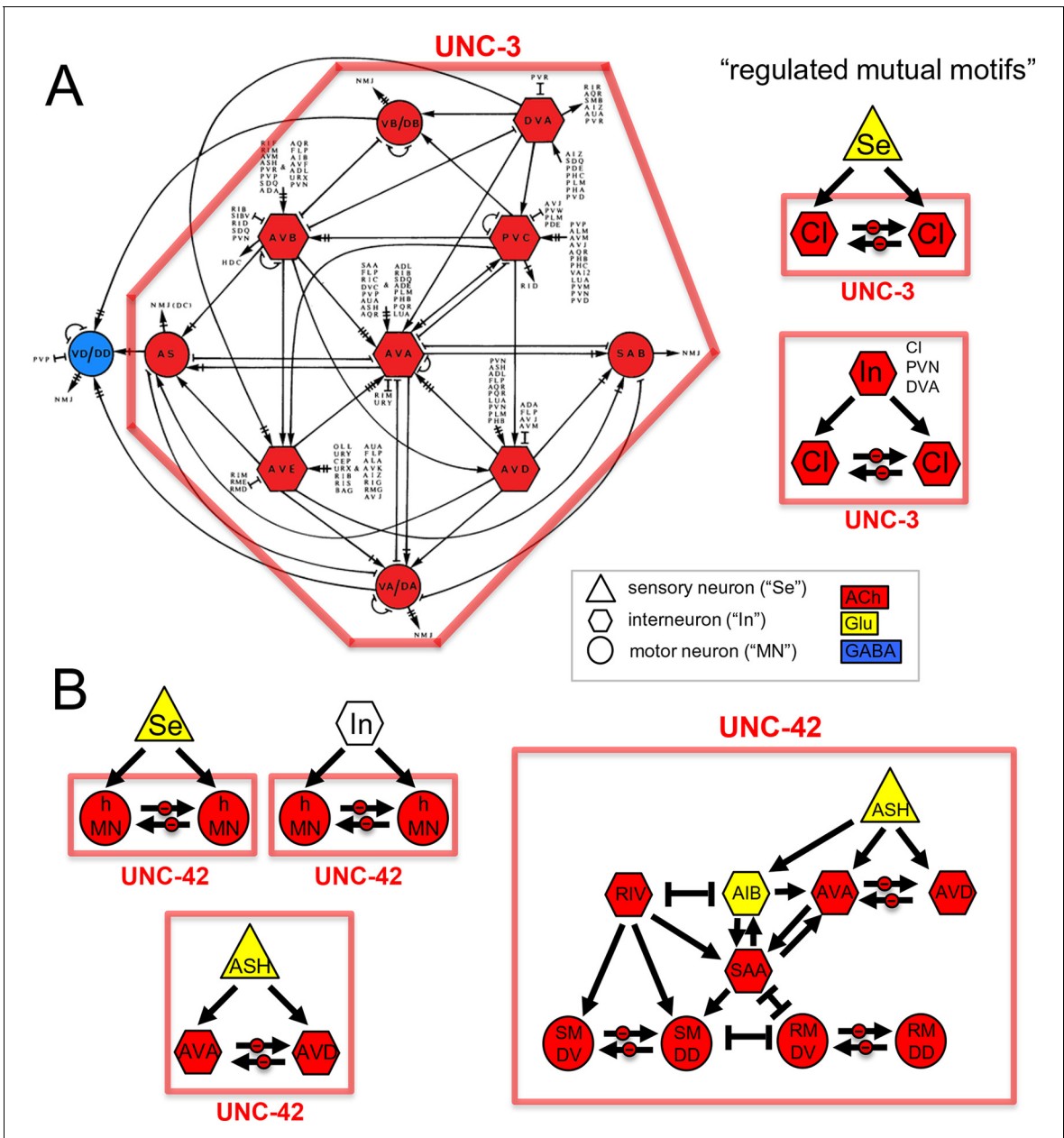

**Figure 9.** Circuit-associated transcription factors. (**A**) Ventral cord motor circuit as shown in *White et al. (1986)*, but now superimposed with neurotransmitter identity and expression pattern of the *unc-3* transcription factor. *unc-3* controls the cholinergic identity of every single neuron in this circuit. Next to the circuit diagram, a number of different regulated mutual 3-neuron networks motifs are shown. These motifs are either embedded in the circuit and provide a connection to neurons located outside the circuit (e.g. glutamatergic sensory neurons). In all cases *unc-3* controls cholinergic identity of the mutually connected command interneurons ('CI') and in those cases where the mutually connected neurons receive cholinergic interneuron input, *unc-3* controls the identity of the entire microcircuit. (**B**) *unc-42* controls the cholinergic identity of interconnected head motor neurons, and glutamatergic signaling between ASH sensory neurons (whose glutamatergic identity is controlled by *unc-42* (*Serrano-Saiz et al., 2013*) and cross-connected command interneurons in which *unc-42* controls glutamate receptor expression (*Brockie et al., 2001*) (shown in *Table 4*). Red boxes indicate the neurons affected by the indicated transcription factor.

the expression of presynaptic neurotransmitter identity and postsynaptic receptor expression in a repulsive reflex circuit (*Figure 9B*).

Notably, the above-mentioned *unc-42*-dependent 'regulated mutual' 3 neuron network motifs are connected to one another, as illustrated in *Figure 9B*. Mutually connected, *unc-42*-dependent head motor neurons are coupled by electrical synapses (*Figure 9B*). Moreover, the *unc-42*-

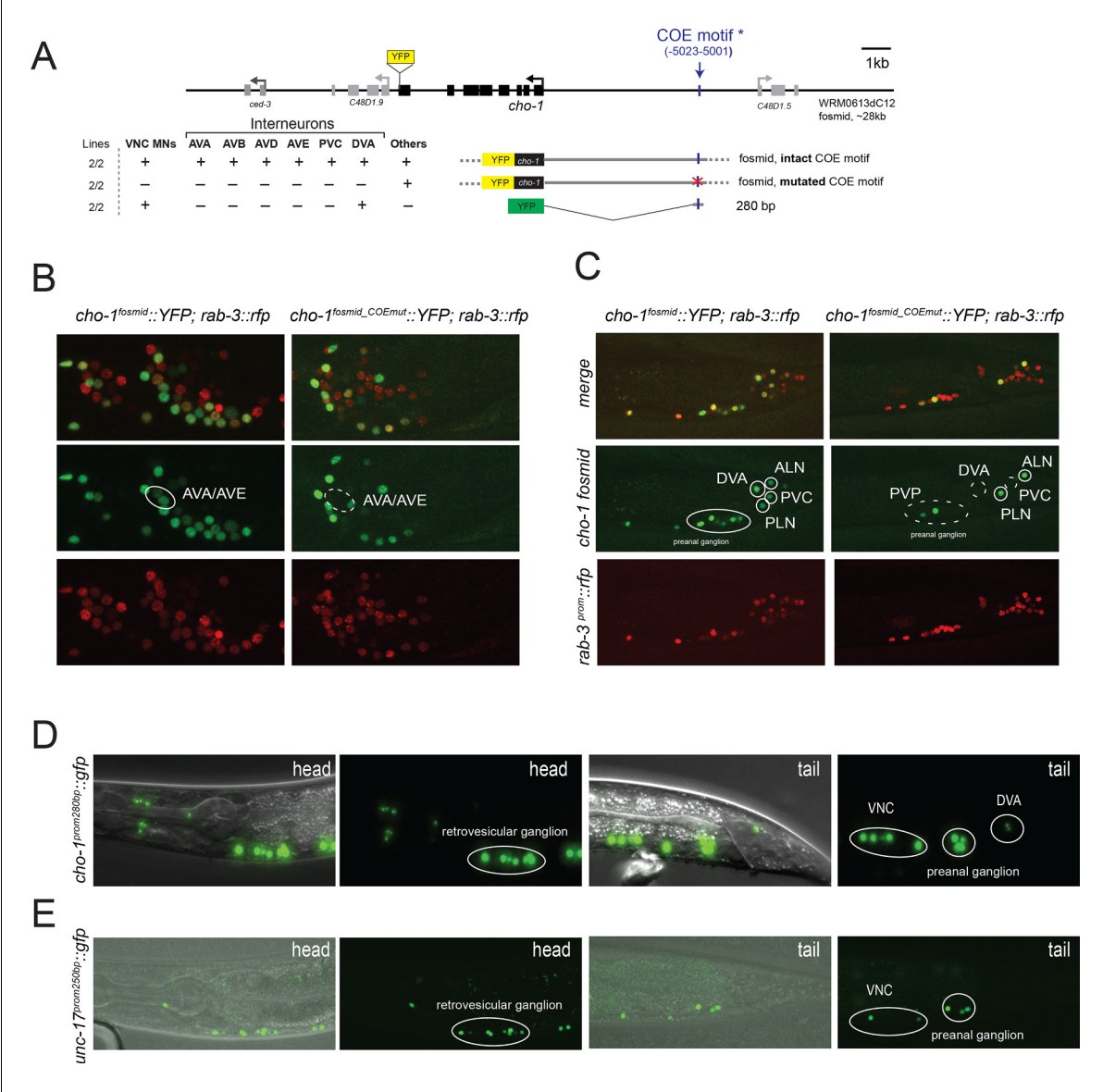

**Figure 10.** A single UNC-3 binding site is required for *cho-1* expression in all distinct *unc-3*-dependent cholinergic neuron types. (**A**) Schematic showing of the *cho-1* locus and the location of the UNC-3 binding site (COE motif) relative to ATG for the fosmid reporters and 280bp promoter fusion. (**B, C**) A *cho-1* fosmid reporter (~28 kb) that contains an intact COE motif shows expression in all cholinergic neurons including the ventral nerve cord (VNC) motor neurons (MNs), the command interneurons (AVA, AVB, AVD, AVE, PVC), and the interneuron DVA. Mutation of the COE motif in the context of this *cho-1* fosmid-based reporter results in selective loss of reporter gene expression in VNC MNs residing at the retrovesicular ganglion and all command interneurons (only AVA and AVE head interneurons are shown). (**B**) *cho-1* fosmid reporter versus *cho-1_COEmut* fosmid reporter in an adult head. (**C**) *cho-1* fosmid reporter versus *cho-1_COEmut* fosmid reporter in an adult tail. Reporter gene expression is also lost in tail neurons DVA and PVC. The transgenic line *rab-3prom::rfp* drives reporter gene expression in the entire nervous system and was used in the background to facilitate neuronal identification. (**D**) A short fragment (280 bp) of the *cho-1 cis*-regulatory region containing the COE motif is sufficient to drive reporter gene expression only in VNC MNs. This fragment does not show expression in command interneurons located at the head and tail of the animal. (**E**) A short fragment (250 bp) of the *unc-17 cis*-regulatory region containing the COE motif is sufficient to drive reporter gene expression only in VNC MNs.

dependent head motor neurons are connected to the *unc-42*-dependent ASH>command interneuron motif. This connection is made by the glutamatergic AIB interneurons; strikingly, their glutamatergic identity is also controlled by *unc-42* (E.S. and O.H., unpubl. data). Taken together, a network of interconnected neurons in the head of the worm that employ distinct neurotransmitter systems all

**Table 9.** Molecular markers for cell identification. The respective markers were crossed with *cho-1* or *unc-17* fosmid reporters to validate cell identification.

| Neuron | Molecular marker |
| --- | --- |
| Hermaphrodite | |
| ADFL/R | *cat-1::GFP (otIs625)* [1] |
| AIA L/R | *ttx-3 fosmid::GFP (wgIs68)* |
| AIN L/R | *ttx-3 fosmid::GFP (wgIs68)* |
| AIY L/R | *ttx-3 fosmid::GFP (wgIs68)* |
| ALN L/R | *unc-86 fosmid::YFP (otIs337); lad-2::GFP (otIs439)* |
| AS1-11 | *unc-3 fosmid::GFP (otIs591)* |
| ASJ L/R | DiI/DiO staining |
| AVA L/R | *glr-1::DsRed (hdIs30); nmr-1::GFP (akIs3)* |
| AVB L/R | *acr-15::GFP (wdEx290); sra-11::GFP (otIs123)* |
| AVD L/R | *glr-1::DsRed (hdIs30); nmr-1::GFP (akIs3)* |
| AVE L/R | *glr-1::DsRed (hdIs30); nmr-1::GFP (akIs3)* |
| AVG | *odr-2::DsRed (otEx4452); unc-6 fosmid::GFP (otEx6370)* |
| AWB L/R | DiI/DiO staining |
| DA1-9 | *unc-3 fosmid::GFP (otIs591); acr-2::GFP (juIs14)* |
| DB1-7 | *unc-3 fosmid::GFP (otIs591); acr-2::GFP (juIs14)* |
| DVA | *ser-2::GFP (otIs358)* |
| HSN L/R | *unc-86 fosmid::YFP (otIs337)* |
| IL2 D/V L/R (x6) | *unc-86 fosmid::YFP (otIs337)* |
| PDA | *unc-3 fosmid::GFP (otIs591); ace-3/4::GFP (fpIs1)* |
| PDB | *unc-3 fosmid::GFP (otIs591)* |
| PLN L/R | *unc-86 fosmid::YFP (otIs337); lad-2::GFP (otIs439)* |
| PVC L/R | *nmr-1::GFP (akIs3)* |
| PVN L/R | [2] |
| PVP L/R | *lin-11 fosmid::GFP (wgIs62); unc-30 fosmid::GFP (wgIs395)* |
| RIB L/R | [2] |
| RIF L/R | *odr-2::DsRed (otEx4452); unc-6 fosmid::GFP (otEx6370)* |
| RIH | *cat-1::GFP (otIs625)* |
| RIR8 | *unc-86 fosmid::YFP (otIs337)* |
| RIV L/R | *unc-42 fosmid::GFP (wgIs173); lad-2::GFP (otIs439)* |
| RMD D/V L/R (x6) | *glr-1::DsRed (hdIs30)* |
| RMF L/R | [2] |
| RMH L/R | [2] |
| SAA D/V L/R (x4) | *lim-4::GFP (mgIs19); lad-2::GFP (otIs439)* |
| SAB D V L/R (x3) | *unc-4::GFP (vsIs45)* |
| SDQ L/R | *unc-86 fosmid::YFP (otIs337); lad-2::GFP (otIs439)* |
| SIA D/V L/R (x4) | *ceh-24::GFP (ccIs4595)* |
| SIB D/V L/R (x4) | *ceh-24::GFP (ccIs4595)* |
| SMB D/V L/R (x4) | *lim-4::GFP (mgIs19); lad-2::GFP (otIs439)* |
| SMD D/V L/R (x4) | *lad-2::GFP (otIs439)* |
| URA D/V L/R (x4) | *unc-86 fosmid::YFP (otIs337)* |
| URB L/R | *unc-86 fosmid::YFP (otIs337)* |
| URX L/R | *flp-10::GFP (otIs92); unc-86 fosmid::YFP (otIs337)* |

*Table 9 continued on next page*

*Table 9 continued*

| Neuron | Molecular marker |
| --- | --- |
| VA1-12 | *unc-3 fosmid::GFP (otIs591); acr-2::GFP (juIs14)* |
| VB1-11 | *unc-3 fosmid::GFP (otIs591); acr-2::GFP (juIs14)* |
| VC1-6 | *lin-11::GFP (nIs106); ida-1::GFP (inIs179)* |
| Pharyngeal | |
| I1 L/R | [3] |
| I3 (L) | [3] |
| M1 (R) | [3] |
| M2 L/R | [3] |
| M4 (L) | [3] |
| M5 (L) | [3] |
| MC L/R | [3] |
| M | |
| CEM D/V L/R (x4) | *pkd-2::GFP (bxIs14)* |
| CA1-9 | *ida-1 (inIs179)* |
| DVE, DVF | [2] |
| HOB | *ida-1 (inIs179)* |
| PCB, PCC, SPC | [4] |
| PDC, PGA | [2] |
| PVV | [2] |
| PVX, PVY | [2] |
| PVZ | *ida-1 (inIs179)* |
| R1A, R2A, R3A, R4A, R6A | [2] |
| SPV | [2] |

[1]Excluded AWA due to lack of overlap of *cho-1* fosmid reporter with *odr-10::gfp* and *unc-17* fosmid reporter *gfp* reporter with *gpa-4::mCherry*. See **Figure 1—figure supplement 1**.

[2]Identified based on position and axonal projections because of the lack of available markers.

[3]Pharyngeal neurons identified based on axonal projections which are visible with the *unc-17* fosmid reporter.

[4]**Garcia et al. (2001)**.

require *unc-42*, either for the acquisition of their neurotransmitter identity or for the ability to receive neurotransmitter signals (*unc-42*-dependent GluR expression in command interneurons).

## Conclusions

- ACh is the most broadly used neurotransmitter in the nematode nervous system. In contrast, glutamate rather than ACh is the most broadly used excitatory neurotransmitter in vertebrates (**von Bohlen Und Halbach and Dermietzel, 2006**). Moreover, while ACh is thought to mainly act as a modulator of other pathways in the vertebrate CNS (**Picciotto et al., 2012**), ACh clearly has a primary role in the transmission and processing of various sensory modalities in *C. elegans*. For example, repulsive odorsensory cues sensed by the AWB olfactory neurons signal exclusively via ACh to command interneuron and motor neurons. The abundant use of ACh by so many different *C. elegans* neuron types may relate to the fact that in invertebrates like *C. elegans*, ACh can not only work as excitatory transmitter, but can also operate as inhibitory neurotransmitter via ACh-gated anion channels (**Putrenko et al., 2005**), which we describe here to be broadly expressed throughout the nervous system. The usage of ACh as an inhibitory neurotransmitter may relate to the fact that apart from its function in motor neurons GABA is only very sparsely used at neuron-neuron-synapses in *C. elegans* (only 1 interneuron).
- The evolutionary history of distinct neurotransmitter systems is still much debated. Sensory-motor neurons can be found in the relatively simple nerve nets of ctenophores (comb jellies)

and possibly form the most primitive and ancient neurons (*Hernandez-Nicaise, 1974*) (Jekely, 2011). The cholinergic nature of most *C. elegans* sensory-motor neurons suggests that ACh was perhaps the first neurotransmitter to have evolved. This is consistent with the notion that ACh is an ancient signaling molecule that precedes the evolution of nervous systems (*Horiuchi et al., 2003*; *Wessler et al., 1999*).

- Sexual identity impinges on the adoption of neurotransmitter identity. One glutamatergic interneuron switches its neurotransmitter to become cholinergic in males. In addition, both hermaphrodite-specific (HSN, VC neurons) and male-specific neurons (CEMs) show a striking delay between birth of the respective neuron and adoption of cholinergic neurotransmitter identity. This delay appears to be independent on sex-specific targets of these neurons.

- We described here seven transcription factors that control cholinergic neurotransmitter identity in 20 cholinergic neuron types. Our findings indicate that combinatorial transcription factor employment patterns the cholinergic nervous system, with individual factors being redeployed in distinct contexts. Moreover, six of the seven regulators of cholinergic identity identified here are homeobox genes. The only non-homeobox gene, *unc-3*, collaborates with one homeobox in two cholinergic neuron types (*ceh-14* PVC and PVN) and possibly more (P.K. and O.H., unpubl. data). Combinatorial employment and homeobox preponderance are also defining features of glutamatergic neurotransmitter identity control (*Serrano-Saiz et al., 2013*).

- Transcription factors that control cholinergic identity also control other identity features of a neuron, suggesting that the principle of co-regulation of many distinct terminal identity features by terminal selector-type transcription factors (*Hobert, 2011*) is broadly employed throughout the nervous system. One exception of this general patterning rule is revealed in the form of UNC-3, which acts as a terminal selector only in VNC MNs, but not in command interneurons. In these neurons, distinct identity features appear to be independently regulated by distinct factors.

- Circuit-associated transcription factors control the neurotransmitter choice of all neurons in synaptically connected cholinergic subcircuits (*unc-3, unc-42*) and/or coordinate the expression of presynaptic neurotransmitter choice and postsynaptic receptor choice (*unc-42*). Cholinergic neurons are also generally heavily interconnected in vertebrate nervous systems (*Woolf, 1991*), and it will be interesting to see whether circuit-associated transcriptional regulators govern their communication as well. For example, vertebrate UNC-3 orthologs (EBF proteins) are not only expressed in cholinergic motor neurons of the spinal cord [(*Garel et al., 1997*); our unpublished observations], but possibly also in cholinergic neurons of the striatum and habenula (*Garel et al., 1999*; *Lobo et al., 2008*; *Nagalski et al., 2015*). Future studies will determine whether EBF proteins act as terminal selectors and/or circuit-associated transcription factors in these neuronal populations.

## Materials and methods

### Mutant strains

The *C. elegans* mutant strains used in this study were: *unc-104(e1265); lim-4(ky403); ceh-14(ch3); lin-11(n389); unc-30(e191); unc-42(e419); unc-86(n846); unc-3(e151); unc-3(n3435)*.

### Transgenic reporter strains

The *unc-17, acc-1, acc-2, acc-3* and *acc-4* fosmid reporter constructs were kindly provided by the TransgeneOme project (*Sarov et al., 2012*). *gfp* is fused directly to the respective loci ('translational reporters'). The *unc-17* fosmid DNA was injected at 15 ng/µl into N2 worms together with *lin-44::yfp* as a co-injection marker. The *cho-1* fosmid reporter construct was generated using λ-Red-mediated recombineering in bacteria as previously described (*Tursun et al., 2009*). For the *cho-1* fosmid reporter, either an SL2 spliced, nuclear- localized mChOpti::H2B sequence was engineered right after the stop codon of the locus (resulting transgene: *otIs544*) or SL2 spliced, nuclear-localized *yfp::H2B* sequence was engineered at the same position, as previously reported (*Stefanakis et al., 2015*) (resulting transgene: *otIs354*). For the 'transcriptional', fosmid-based *unc-7* reporter an *sl2::H2B::yfp* cassette was inserted at the 3' end of the locus. The *acc-1, -2, -3*, and *-4* and the *cho-1* fosmid DNA were injected at 15 ng/µl into a *pha-1(e2123)* mutant strain with pBX as co-injection marker (*Granato et al., 1994*). The following reporter strains were generated for this study: *unc-17*

fosmid reporter (*otIs576*), *cho-1* fosmid reporter (*otIs544*), *acc-1* fosmid reporter (*otEx6374*), *acc-2* fosmid reporter (*otEx6375*), *acc-4* fosmid reporter (*otEx6376*). For the mutation of the COE motif in the context of the *cho-1* fosmid-based reporter construct, two nucleotides (wild-type COE motif: aaaacggtct**cc**agggagagaaa; mutated COE motif: aaaacggtct**gg**agggagagaaa) that are critical for UNC-3 binding were mutated as previously described in *Stefanakis et al., 2015*.

The following additional, and previously described neuronal markers were used in the study: *ric-19::gfp (otIs380)*, *eat-4^fosmid^::sl2::yfp::H2B (otIs388)*, *eat-4^fosmid^::sl2::mChOpti::H2B (otIs518)*, *ace-3/4::gfp (fpIs1)*, *rab-3::bfp (otIs355)*, *pkd-2::gfp (bxIs14)*, *unc-86^fosmid^::yfp (otIs337)*, *rab-3::rfp (otIs355)*, *lin-39^fosmid^::gfp (wgIs18)*, *opt-3::gfp (gvEx173)*, *flp-18::TagRFP (otEx6491)*, *rig-3::gfp (otEx239)*. Additional transgenes used for cell identifications are listed in *Table 9*.

## Cell-type specific changes of sexual identity

FEM-3 and TRA-2ic were expressed under the control of a fragment of the *eat-4* locus (between -2680 and -2155bp from ATG). The plasmids were injected in *him-5(e1490)* at 50 ng/ul. Two lines expressing FEM-3 were then crossed with *otIs388 (eat-4^fosmid^::sl2::yfp::H2B)* and *otIs354 (cho-1^fosmid^::sl2::yfp::H2B)* independently generating the following strains: OH13753 [*otIs388; otEx6377 (eat-4^prom11^::fem-3::sl2::tagRFP;unc-122::GFP)*], OH13802 [*otIs388; otEx6378 (eat-4^prom11^::fem-3::sl2::tagRFP;unc-122::GFP)*], OH13805 [*otIs354; otEx6377 (eat-4^prom11^::fem-3::sl2::tagRFP;unc-122::GFP)*] and OH13806 [*otIs354; otEx6378 (eat-4^prom11^::fem-3::sl2::tagRFP;unc-122::GFP)*]. Similarly, two lines expressing TRA-2ic were crossed with *otIs388* and *otIs354* generating the following strains: OH13803 [*otIs388; otEx6379 (eat-4^prom11^::tra-2ic::sl2::tagRFP; unc-122::GFP)*], OH13804 [*otIs388; otEx6380 (eat-4^prom11^::tra-2ic::sl2::tagRFP; unc-122::GFP)*], OH13807 [*otIs354; otEx6379 (eat-4^prom11^::tra-2ic::sl2::tagRFP; unc-122::GFP)*] and OH13808 [*otIs354; otEx6380 (eat-4^prom11^::tra-2ic::sl2::tagRFP; unc-122::GFP)*].

*eat-4* and *cho-1* expression was analyzed at 1 day adult animals with and without the array. *eat-4* expression in AIM was normalized by its expression in the RIGL/R neurons, while for *cho-1* the expression in AIM was normalized by its expression in AIYL/R neurons.

## Antibody staining

Immunofluorescence for UNC-17 was performed as described earlier (*Duerr et al., 2008*) using a an *unc-104(e1265)* mutant strain carrying the *cho-1* fosmid reporter (*otIs544*). Worms were fixed using methanol/acetone and stained with a rabbit anti-UNC-17 serum diluted 1/100 (gift from James Rand).

Immunofluorescence for serotonin was performed using a tube fixation protocol as described earlier (*Serrano-Saiz et al., 2013*). The anti-5HT S-5545 antibody was used at 1/100 and purchased from Sigma Aldrich.

## Microscopy

Worms were anesthetized using 100 mM of sodium azide (NaN$_3$) and mounted on 5% agarose on glass slides. All images (except *Figure 8* and *Figure 7—figure supplement 1*) were acquired using a Zeiss confocal microscope (LSM880). Several z-stack images (each ~0.4 µm thick) were acquired with the ZEN software. Representative images are shown following orthogonal projection of 2–10 z-stacks. Images shown *Figure 8* and *Figure 7—figure supplement 1* were taken using an automated fluorescence microscope (Zeiss, AXIO Imager Z1 Stand). Acquisition of several z-stack images (each ~1 µm thick) was performed with the Micro-Manager software (Version 3.1). Representative images are shown following max-projection of 2–10 z-stacks using the maximum intensity projection type. Image reconstruction was performed using ImageJ software (*Schneider et al., 2012*).

For quantification of UNC-7::GFP puncta shown in *Figure 7C*, images were acquired and z-stack were generated as described above. Manual counting of the UNC-7::GFP puncta was performed using the cell counter plug-in of the ImageJ software.

For the quantification of *eat-4* and *cho-1* expression in AIM for the analysis shown in *Figure 5F*, images were acquired using a Zeiss confocal microscope (LSM880) and the fluorescence intensity mean was obtained with the ZEN software tool.

## Statistical analysis

For results shown in *Figures 5E–F and Figure 7C* statistical analysis was performed using the Student's *t*-test (tail 2, type 2). Values are expressed as mean ± standard deviation (s.d.) or standard error of the mean (sem) as indicated in each figure legend. For results shown in *Figure 7* and *Figure 7—figure supplement 1* we performed Fisher's exact test (two-tailed).

## Electron microscopical analysis

Wild-type and *unc-3* mutant animals were reconstructed in the anterior region of the ventral cord in order to determine neuron morphology and synaptic circuitry. The reconstructions were made from electron micrographs of serial sections as described in *White et al. (1986)*. The regions reconstructed were ~150 μm in length and included ~1800 serial sections. We reconstructed the region of the ventral nerve cord that roughly includes the region from AS01 to AS03 motor neurons. Every third section was photographed and printed. All the processes of neurons with cell bodies in the region reconstructed were followed. The neurons were identified by characteristic synaptic or morphological features together with the relative position of their cell bodies in the sequence of cell bodies in the ventral cord (*White et al., 1986*). The two reconstructed animals were the wild-type N2U and the *unc-3*(*e151*) allele in *trans* to a covering deficiency (*mnDf5*). This strain was generated by crossing *unc-3*(*e151*) the strain SP266 mnDp1(X;V)/ V; mnDf5 X.

## Screen for cholinergic identity mutants

The *otIs341 (mgl-1::gfp)* transgenic strain was used to identify mutants affecting the identity of the cholinergic RMDD/V motor neurons. A conventional semi-clonal EMS screen identified the *ot712* mutation, which was found to be closely linked to the transgene *vsIs33V* also present in the strain background. *ot712* animals are uncoordinated and *unc-42* maps on LGV. Complementation tests between *ot712* and two alleles of *unc-42* (*e419* and *e270*) confirmed that *ot712* is an allele of *unc-42*. Sanger sequencing reveals that *ot712* harbors a late nonsense mutation (W181>Stop) in exon 6.

## Network analysis

### Network Construction

Connectivity data was taken from the latest release of www.wormwiring.org which contains updates to the original wiring diagram (*White et al., 1986*). Data from both JSE and N2U worms were used. Only connections that have more than 3 EM-serial sections of synaptic connection are kept. Connection between a neuron to itself were ignored.

### Motifs identification

We used m-finder software (located at: http://www.weizmann.ac.il/mcb/UriAlon/download/network-motif-software) to find 3-neuron and 4-neuron network motifs. 3-neuron motifs analysis was performed with default parameters. 4-motifs search was performed with the metropolis randomization method

### Neurotransmitter assignment and enrichment analysis

We used five NT categories: Glutamate, Acetylcholine, GABA, other, Unknown. A neuron secreting GABA/Glu/Ach another NT was mapped as GABA/Glu/ACh. A neuron secreting only NTs other than GABA/Glu/Ach was mapped as 'Other'. All neurons without a NT were named 'Unknown'.

Motif role: A motif-role is a subset of neurons in a motif that when interchanged the subgraph remains the same motif. For example, for the 'regulated mutual motif' there are 2 roles: one role that comprises of neurons [1, 3] and one role that comprises of neuron [2].

We employ two approaches in search for NT-enrichment:

1. Per motif-role, we ask if neurons in this role secrete a specific NT more often than expected by chance.
2. For motif-roles that comprises of more than one NT, we further ask which NT-combinations appear more than expected by chance.

In both approaches, we start by counting the events in the real network. In approach 1, we count for each motif role and a given NT how many neurons in that motif role secrete that specific NT in

the real neuron-network. For approach number 2, we count the number of occurrences of each NT-combination per given role.

We then compare those numbers to the event-numbers in randomized networks. Each generated network has the same network-structure, but each neuron is associated with a random NT out of the NT pool. This is done without repetitions, so at the end the randomized NT-list is a shuffling of the real NT-list. We generate 1000 randomized networks. We count for each randomized network how many times each NT appeared in each motif-role (for approach 1) and how many times each NT-combination appeared at each motif-role (approach 2). The number of occurrences of each NT/NT-combination is compared between the real network and the randomized networks by calculating a z-score.

$$zscore = \frac{\#occurences\,in\,real\,network - Mean(\#occurences\,in\,randomized\,networks)}{STD(\#occurences\,in\,randomized\,networks)}$$

We say that a motif-role is significantly associated with a given NT (approach 1), or a NT-combination (approach 2) if the z-score is larger than 2.

To assess the risk for false-positives in this approach, we repeated the above calculation while treating 100 randomized networks as real networks, comparing each of them to 100 randomized networks, and choosing events with z-score >2 as significant. For 3-neuron motifs, approach 1, the real network has 5.6 ± 0.7 significant events (the average was done by each time comparing the real network to different randomized networks), while random networks had on average 1.8 ± 1.9 significant events. Hence, the risk for false-positives does not seem high.

## Calculating the relative usage of neurotransmitters as a function of the processing depth

We obtained the processing depth of each neuron from *Varshney et al. (2011)*. Then, for each neurotransmitter, we calculated a histogram of the number of neurons that use this neurotransmitter at each processing depth (using the function SmoothHistogram in Mathematica). We then normalized the graphs at each processing depth such that each graph represents the percentage of neurons that use the relevant neurotransmitter at the relevant processing depth.

## Acknowledgements

We thank Qi Chen for generating transgenic strains, Feifan Zhang for isolating and providing the *ot712* allele, James Rand for the UNC-17 antibody, Tulsi Patel for *gfp*-tagging the endogenous *unc-3* locus, Nikos Stefanakis for engineering the COE motif mutation in the *cho-1* fosmid, Abhishek Bhattcharya for *unc-7* reporters, Eileen Southgate, Nichol Thomson for help in EM reconstructions of *unc-3*, Benjamin Raja for help with the EM analysis of *unc-3* mutants, Scott Emmons for discussion and providing data via www.wormwiring.org, Kelsey Roberts for helping with genetic analysis and Shawn Lockery for communicating unpublished results. We are very grateful to James Rand for much helpful advice during this study and for detailed comments on the manuscript. We also thank Shawn Lockery and Christoph Kellendonk and members of the Hobert lab for comments on the manuscript. This work was funded by the National Institutes of Health [R37NS039996-05 to OH, K99-NS-084988-02 to PK), OD 010943 to DHH] and the Howard Hughes Medical Institute (LRG and OH).

## Additional information

### Competing interests
OH: Reviewing editor, *eLife*. The other authors declare that no competing interests exist.

### Funding

| Funder | Author |
|---|---|
| Howard Hughes Medical Institute | Laura Pereira Oliver Hobert |

National Institute of Neurological Disorders and Stroke

Oliver Hobert

The funders had no role in study design, data collection and interpretation, or the decision to submit the work for publication.

## Author contributions

LP, Conception and design, Acquisition of data, Analysis and interpretation of data, Drafting or revising the article; PK, ESS, Acquisition of data, Analysis and interpretation of data, Drafting or revising the article; HS, BL, LRG, Analysis and interpretation of data; AEM, DHH, JGW, OH, Acquisition of data, Analysis and interpretation of data; UA, Conception and design, Analysis and interpretation of data, Drafting or revising the article

## Author ORCIDs

Avi E Mayo, http://orcid.org/0000-0002-4479-3423
David H Hall, http://orcid.org/0000-0001-8459-9820
Oliver Hobert, http://orcid.org/0000-0002-7634-2854

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
