## [Decision Letter]

Thank you for submitting your work entitled "A cellular and regulatory map of the cholinergic nervous system of C. elegans" for consideration by *eLife*. Your article has been reviewed by a Senior Editor, Kang Shen (Reviewing Editor) and two peer reviewers. The following individuals involved in review of your submission have agreed to reveal their identity: Gerald Rubin and Shawn Xu (peer reviewers).

The reviewers have discussed the reviews with one another and the Reviewing Editor has drafted this decision to help you prepare a revised submission.

Summary:

Pereira et al., have written a very interesting and highly informative article which will be extremely beneficial for the *C. elegans* community. This article sets a broad foundation for cholinergic signaling in the *C. elegans* nervous system. They use an array of imaging techniques to identify potential sites of release, reception and cholinergic neuron identities, as well as acute observation and discussion of the role of acetylcholine in circuit motifs.

Both reviewers thought that the manuscript is suitable for publication in *eLife* after the following questions are addressed.

Essential revisions:

1) It would be helpful to readers to expand on the generality of the statement that "Invertebrate genomes encode not only conventional excitatory Ach-gated cation ion channels, but also inhibitory Ach-gated anion ion channels". For example, is there evidence that the widely studied invertebrate *Drosophila* has inhibitory nAChRs? In particular, Wormbase lists the closest fly homolog of ACC-4 as the GABA receptor Rdl. If evidence for the existence of Ach-gated anion ion channels is limited to *C. elegans*, then perhaps the wording should be changed to say "the *C. elegans* genome encodes" to be consistent with the available facts. I note that the two cited references do not contain information to support the broader statement made, as they describe work with *C. elegans* only.

2) I think the statement in paragraph seven subsection “*unc-3* is a circuit-associated transcription factor” that *unc-3* activity is required to "specify" synaptic connectivity is too strong and misleading. The data just show that *unc-3* activity is required for normal synaptic connectivity to occur. That means something very different, at least to me, than "specify". Likewise the statement that *unc-3* "defines the function and assembly of an entire circuit" goes way beyond what can be concluded from the results presented. The circuit is indeed abnormal in *unc-3* mutants, but to use the word "define" here would require, in my opinion, evidence that in *unc-3* mutants the circuit became a different type of functional circuit, not that it is just abnormal. I think one can say that *unc-3* is "required", but "define" implies a specific role for *unc-3* that has not been demonstrated. Similarly "circuit-associated" is appropriate, but "circuit-defining" is not.

3) This present study sets a foundation for investigating the role of ACC-4 in cholinergic signaling. ACC-4 however does not elicit strong Ach-gated chloride currents when expressed alone (Putrenko et al., 2005), does this suggest that it is an auxiliary subunit in a heteropentameric receptor? Therefore, does *acc-4* co-express with at least one other Ach-gated chloride channel in each cell?

---

## [Author Response]

*1) It would be helpful to readers to expand on the generality of the statement that "Invertebrate genomes encode not only conventional excitatory Ach-gated cation ion channels, but also inhibitory Ach-gated anion ion channels". For example, is there evidence that the widely studied invertebrate Drosophila has inhibitory nAChRs? In particular, Wormbase lists the closest fly homolog of ACC-4 as the GABA receptor Rdl. If evidence for the existence of Ach-gated anion ion channels is limited to* C. elegans*, then perhaps the wording should be changed to say "the* C. elegans *genome encodes" to be consistent with the available facts. I note that the two cited references do not contain information to support the broader statement made, as they describe work with* C. elegans *only.*

We apologize for this misstatement. While *Drosophila* does have anion-gated LGICs, it’s not clear whether these are ACh channels; in fact, the specific clade of ACC genes in worms is NOT conserved in *Drosophila*. So, while we can’t rule out that they exist in flies, there is currently no evidence for this. We followed the reviewer’s suggestion and changed the sentence to "the *C. elegans* genome encodes…"

*2) I think the statement in paragraph seven subsection “*unc-3 *is a circuit-associated transcription factor” that* unc-3 *activity is required to "specify" synaptic connectivity is too strong and misleading. The data just show that* unc-3 *activity is required for normal synaptic connectivity to occur. That means something very different, at least to me, than "specify". Likewise the statement that* unc-3 *"defines the function and assembly of an entire circuit" goes way beyond what can be concluded from the results presented. The circuit is indeed abnormal in* unc-3 *mutants, but to use the word "define" here would require, in my opinion, evidence that in* unc-3 *mutants the circuit became a different type of functional circuit, not that it is just abnormal. I think one can say that* unc-3 *is "required", but "define" implies a specific role for* unc-3 *that has not been demonstrated. Similarly "circuit-associated" is appropriate, but "circuit-defining" is not.*

Agree with all, fixed as requested (i.e. eliminated “specify” and “defining”).

*3) This present study sets a foundation for investigating the role of ACC-4 in cholinergic signaling. ACC-4 however does not elicit strong Ach-gated chloride currents when expressed alone (Putrenko et al., 2005), does this suggest that it is an auxiliary subunit in a heteropentameric receptor? Therefore, does* acc-4 *co-express with at least one other Ach-gated chloride channel in each cell?*

Yes, that’s correct. Unpublished worked from another *C. elegans* lab has indeed found such a subunit (an *lgc* gene). Since this is unpublished data, we have not mentioned this in the manuscript.